# P2X7 receptor inhibition ameliorates dendritic spine pathology and social behavioral deficits in Rett syndrome mice

Juan Mauricio Garré[1,2 ✉], Hernandez Moura Silva[3], Juan J. Lafaille[3,4] & Guang Yang [1,2 ✉]

Dysregulated immunity has been implicated in the pathogenesis of neurodevelopmental disorders but its contribution to synaptic and behavioral deficits in Rett syndrome (RTT) remains unknown. P2X7 receptors (P2X7Rs) are unique purinergic receptors with pro-inflammatory functions. Here, we report in a MECP2-deficient mouse model of RTT that the border of the cerebral cortex exhibits increased number of inflammatory myeloid cells expressing cell-surface P2X7Rs. Total knockout of P2X7Rs in MECP2 deficient mice decreases the number of inflammatory myeloid cells, restores cortical dendritic spine dynamics, and improves the animals' neurological function and social behavior. Furthermore, either genetic depletion of P2X7Rs in bone-marrow derived leukocytes or pharmacological block of P2X7Rs primarily outside of the central nervous system parenchyma, recapitulates the beneficial effects of total P2X7R depletion on the social behavior. Together, our results highlight the pathophysiological roles of P2X7Rs in a mouse model of RTT.

[1] Department of Anesthesiology, Columbia University Medical Center, New York, NY 10032, USA. [2] Department of Anesthesiology, New York University School of Medicine, New York, NY 10016, USA. [3] Kimmel Center for Biology and Medicine at the Skirball Institute, New York University School of Medicine, New York, NY 10016, USA. [4] Department of Pathology, New York University School of Medicine, New York, NY 10016, USA. ✉email: jmg2340@cumc.columbia.edu; gy2268@cumc.columbia.edu

Rett syndrome (RTT) is a postnatal neurodevelopmental disorder caused by mutations in the X-linked gene encoding methyl-CpG-binding protein 2 (MECP2). Although rare, RTT is the most frequent cause of developmental disability in girls, occurring in 1 of 10,000–15,000 births. In males, RTT is less prevalent, but more severe[1,2]. There is no cure yet for Rett syndrome. Current treatments aim to ameliorate the patients' neurological symptoms and improve their communication skills and social engagement. Rett syndrome has been modeled in rodents with loss of function mutations in MECP2. These mice develop neurological and behavioral phenotypes resembling those observed in RTT patients. The main function of MeCP2 protein is to regulate the transcription of methylated genes[3–5]. Although the lack of neuronal MeCP2 in the central nervous system (CNS) is thought to account for the majority of symptoms associated with RTT, MeCP2 deficiency outside of the CNS may also contribute to some aspects of RTT-like neurological dysfunction in mice[6,7].

Increasing evidence has linked MECP2 mutations and dysregulated immunity. For example, loss of function mutations in MECP2 have been shown to alter the immune cell response to lipopolysaccharides[8]. MECP2 gain of function mutations are associated with decreased immune responses in Th1 lymphocytes and natural killer cells, as well as higher monocyte counts in blood[9]. Individuals with MECP2 loss or gain of function mutations display a higher prevalence of unresolved inflammation and infections[9–11]. In addition, MECP2 deficiency leads to hyperactivity in hypothalamus pituitary axis and elevation of stress hormone levels in blood[12], both of which have been associated with neurodevelopmental disorders[13]. These conditions may affect leukocyte function, leading to synaptic and behavioral alterations[14]. Nevertheless, whether and to what extent dysregulated peripheral immunity contributes to brain dysfunction and RTT symptoms is unknown.

P2X7 receptors (P2X7Rs) are a class of unique ligand-gated nonselective channels and are broadly expressed in immune cells, particularly in cells of myeloid lineage[15]. In CNS, P2X7Rs have been mainly detected in glial cells and there is no consensus on neuronal expression of P2X7Rs[16]. Upon activation by ATP, P2X7Rs regulate the production and release of inflammatory mediators, such as IL-1β, TNFα, and PGE2, as well as the oxidation of nitric oxide derivates[17,18]. Given their proinflammatory function, P2X7Rs have been proposed as therapeutic targets for various inflammatory and neurological disorders[15,17,19–21] as well as for autism-like behavior in mice with maternal immune activation[22–24].

In this study, we investigated the pathophysiological roles of P2X7Rs in mice expressing a truncated form of MeCP2 (Mecp2$^{308/Y}$, also known as Mecp2$^{tm1Hzo}$), which is a well-established model to study RTT phenotypes in male mice, avoiding the variability of behavioral outcomes linked to X-chromosome inactivation[25]. We showed that MECP2 deficiency resulted in an accumulation of P2X7R-expressing monocytes and macrophages in the border of the cerebral cortex. P2X7R deficiency in Mecp2$^{308/Y}$ mice reduced the number of these immune cells in the brain, restored cortical dendritic spine plasticity, and ameliorated social behavioral defects. Importantly, either genetic ablation of P2X7Rs in peripheral leukocytes or pharmacological inhibition of P2X7Rs primarily outside of the CNS parenchyma partially recapitulated the beneficial effects of total P2X7R depletion. Together, our results underscore the contributions of P2X7Rs and non-microglial myeloid cells to brain dysfunction in RTT.

## Results

**Mecp2$^{308/Y}$ mice have more cortical monocytes and macrophages.** To investigate the effects of MECP2 deficiency on immune cells in the brain, we performed immunohistochemistry using antibodies targeting either myeloid cells broadly (Iba1) or nonmicroglial macrophages alone (CD206). With the integrity of perivascular spaces preserved during perfusion[26], we observed numerous cells expressing Iba1 or CD206 along the border of the cerebral cortex of Mecp2$^{308/Y}$ mice with neurological dysfunction (Fig. 1a). As compared to age- and sex-matched WT mice, Mecp2$^{308/Y}$ mice had a higher number of Iba1$^+$ cells in the border of the cerebral cortex (defined as the brain region within 30 μm from pial surface, including pia mater), but not in cortical parenchyma >30 μm away from pial surface (Fig. 1a–c). Moreover, there were more CD206$^+$ cells in the cortical border of Mecp2$^{308/Y}$ than WT mice (Fig. 1a, d). Iba1$^+$ and CD206$^+$ cells were also observed in prominent perivascular spaces of the large penetrating cortical vessels (Fig. 1a). Consistent with previous reports[27], we did not detect CD206$^+$ cells in cortical parenchyma in both WT and Mecp2$^{308/Y}$ mice.

To further characterize the leukocyte populations in cortex, we performed multiparametric flow cytometry, focusing on the myeloid subsets (CD45$^+$CD11b$^+$) (see Supplementary Tables 1 and 2 for leukocyte cell surface markers). We found that the prevalence of Ly6C$^{High}$ monocytes within CD45$^+$ (leukocyte common antigen) cells was higher in the cortex of Mecp2$^{308/Y}$ mice as compared to WT cohorts (Fig. 1e, f). By gating out monocytes (Ly6C$^−$) and neutrophils (Ly6G$^−$) and analyzing the expression of CD38 and MHCII in the remaining CD64$^+$ macrophages, we were able to identify three major myeloid subsets in the mouse cortex: microglia (CD38$^−$MHCII$^{Low}$), CD38$^+$MHCII$^{Low}$ and CD38$^−$MHCII$^{High}$ macrophages (Supplementary Fig. 1a). Further characterization of these subsets showed that microglia were CD206$^{Low}$MHCII$^{Low}$, whereas CD38$^+$MHCII$^{Low}$ and CD38$^−$MHCII$^{High}$ macrophage populations were CD206$^{High}$MHCII$^{Low}$ and CD206$^{Low}$MHCII$^{High}$, respectively (Fig. 1g). Both CD206$^{High}$MHCII$^{Low}$ and CD206$^{Low}$MHCII$^{High}$ macrophages are thought to be enriched in nonparenchymal brain tissues under physiological conditions, and based on their anatomical distribution, these populations have been recently referred to as vascular-associated macrophages in the adipose tissue[28] and perivascular- and CNS-associated or border-associated macrophages in the brain[27,29,30]. We found that CD206$^{High}$MHCII$^{Low}$ and CD206$^{Low}$MHCII$^{High}$ macrophages increased their frequency by twofolds in the cortical border of Mecp2$^{308/Y}$ mice as compared to WT cohorts, whereas microglia remained unaltered (Fig. 1g–j). There were no changes in the frequency of myeloid cells in the subcortical regions and cerebellum of Mecp2$^{308/Y}$ mice (Supplementary Fig. 1b, c). In cortex, there were no significant differences in the fractions of Ly6C$^{Low}$ monocytes and neutrophils between WT and Mecp2$^{308/Y}$ mice (Supplementary Fig. 1d). In blood, there were fewer CD4$^+$ lymphocytes and altered Ly6C$^{High}$/Ly6C$^{Low}$ monocyte fractions in Mecp2$^{308/Y}$ as compared to WT mice (Supplementary Fig. 1e, f).

Microglia are important for neurodevelopment[31], and both activation and loss of microglia have been observed in Mecp2$^{tm1.1Bird}$ mice, another mouse model of RTT[8]. Notably, in the cortex of Mecp2$^{308/Y}$ mice, we did not observe any morphological signs of microglial activation (Fig. 1k, l), although the density of microglia was reduced by 10% as compared to that in WT mice (Fig. 1c). There were no differences in the levels of myeloid activation markers MHCII and CD11c in microglia between WT and Mecp2$^{308/Y}$ mice (Fig. 1m, n).

In summary, we observed increased density (by immunohistochemistry) and frequency (by flow cytometry) of monocytes and nonmicroglial macrophages in the brain of Mecp2$^{308/Y}$ mice, suggesting a predisposed inflammatory phenotype as a result of MECP2 deficiency.

**Monocytes and macrophages express cell surface P2X7Rs.** P2X7Rs are broadly expressed by myeloid cells[15]. We analyzed

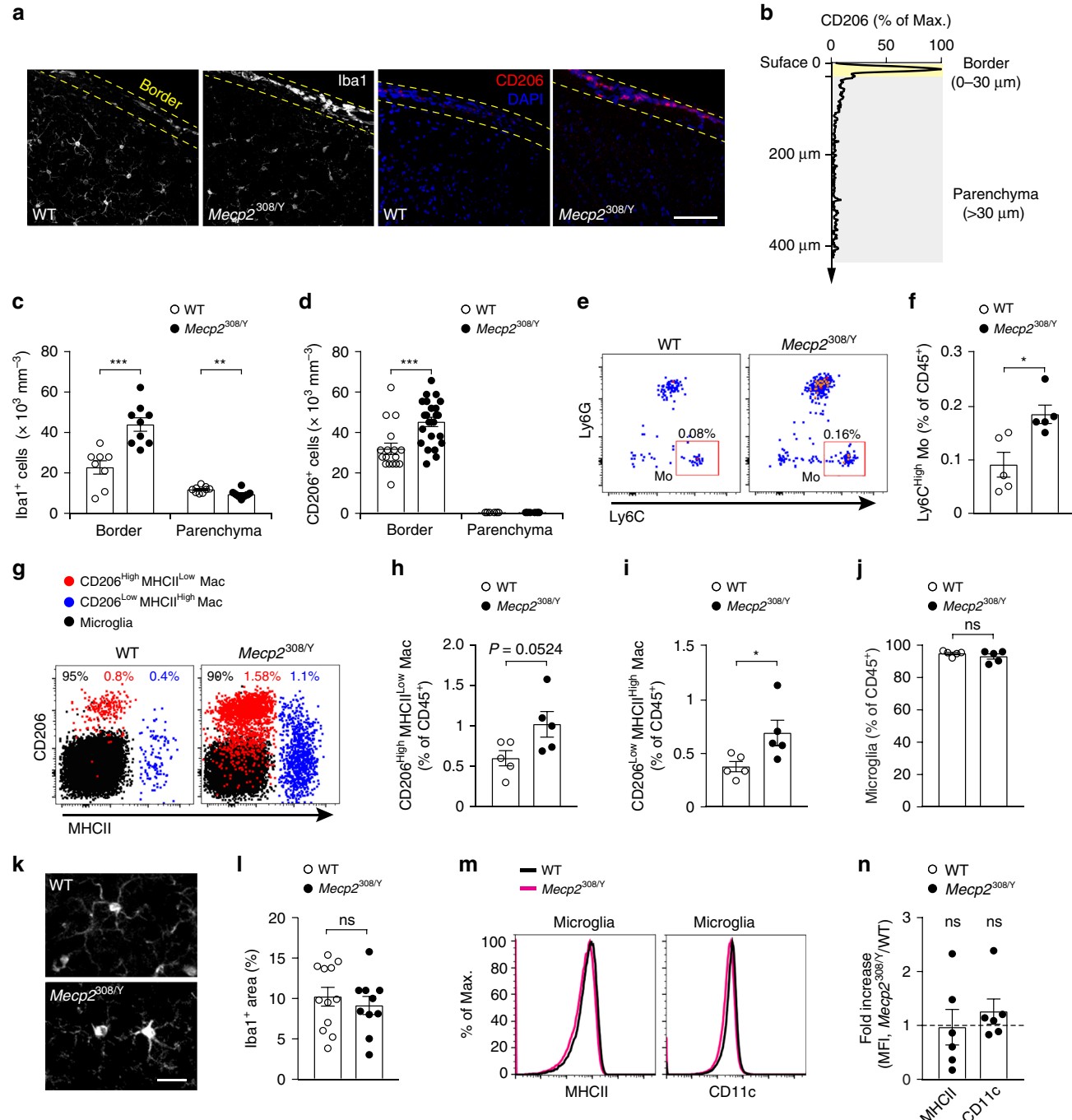

**Fig. 1 *Mecp2*[308/Y] mice have more monocytes and macrophages in the cortical border. a** Representative coronal sections of the mouse cortex stained for Iba1 and CD206 in WT and *Mecp2*[308/Y] mice (4–6 months old). Dotted yellow lines indicates the border region (within 30 μm from pial surface) of the cerebral cortex. Scale bar, 80 μm. **b** Representative distribution of CD206 fluorescence intensity across the depth of the cortex. CD206 expression was mainly observed in the border region of the cortex. **c** Quantification of Iba1[+] cell density in the cortical border ($n = 8, 9$; $P = 0.0004$) and parenchyma ($n = 9, 10$; $P = 0.0096$). Empty circle, WT; Filled circle, *Mecp2*[308/Y]. **d** Quantification of CD206[+] cell density in the cortical border ($n = 17, 23$; $P = 0.0007$) and parenchyma ($n = 6, 6$). **e** Representative flow cytometry analysis showing the percentages of Ly6C[High] monocytes (Mo) in the cortex of WT and *Mecp2*[308/Y] mice. **f** Quantification of data shown in (**e**) ($n = 5, 5$; $P = 0.0113$). Monocytes were gated from CD45[High]CD11b[+]MHCII[−]Ly6G[−] cells. **g** Representative flow cytometry analysis showing surface expression of CD206 and MHCII in microglia and macrophages (Mac) of WT and *Mecp2*[308/Y] mice. **h–j** Percentages of CD206[High]MHCII[Low] macrophages ($P = 0.0524$), CD206[Low]MHCII[High] macrophages ($P = 0.0393$), and microglia ($P = 0.3278$) in the cortex of WT and *Mecp2*[308/Y] mice ($n = 5$ mice per group). Relative numbers are expressed as percentages of CD45[+] cells. **k** Representative images of microglia in the cortex of WT and *Mecp2*[308/Y] mice. Scale bar, 25 μm. **l** Quantification of area covered by Iba1[+] microglia ($n = 12, 10$; $P = 0.5123$). **m** Histograms showing distribution of surface MHCII and CD11c expression in cortical microglia in WT or *Mecp2*[308/Y] mice. **n** Quantification of data shown in **m** ($n = 6$ per group, $P = 0.9178$ and 0.3032). Data are presented as mean ± SEM. *$P < 0.05$, **$P < 0.01$, ***$P < 0.001$, ns not significant, unpaired two-tailed *t* test was used in (**c**, **d**, **f**, **h–j**, **l**, **n**). The source data underlying **c**, **d**, **f**, **h–j**, **l**, **n** are provided as a Source Data file.

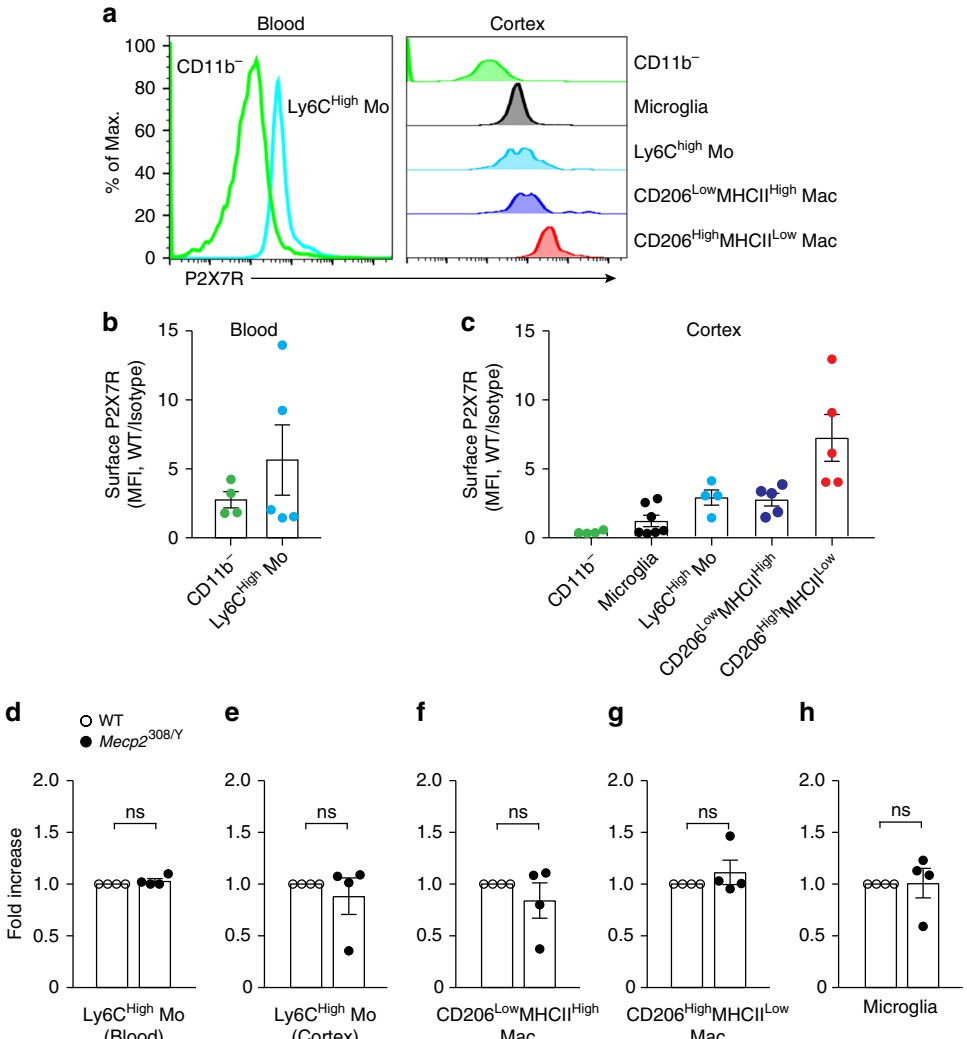

**Fig. 2 P2X7Rs are abundantly expressed on monocytes and macrophages. a** Histograms showing distribution of surface P2X7R expression in lymphocytes and Ly6C[High] monocytes in blood, and lymphocytes, microglia, monocytes (Mo) and macrophages (Mac) in the WT cortex. **b**, **c** Quantification of data shown in (**a**) in blood (**b**; $n = 4$, 5 mice) and cortex (**c**; $n = 4, 7, 4, 5, 5$). The median fluorescence intensity (MFI) of cells stained with an anti-P2X7R antibody is normalized to the isotype IgG negative controls. P2X7R[Low], P2X7R[int], and P2X7R[High] are defined as <2, 2–3, and >3 in MFI. **d–h** Surface P2X7R expression in blood Ly6C[High] monocytes **d**, cortical Ly6C[High] monocytes **e**, CD206[Low]MHCII[High] macrophages **f**, CD206[High]MHCII[Low] macrophages **g**, and microglia **h** of Mecp2[308/Y] relative to WT mice ($n = 4$ mice per group). Empty circle, WT; filled circle, Mecp2[308/Y]. Unpaired two-tailed $t$ test was used in (**d–h**). Summary data are presented as mean ± SEM. The source data underlying **b–h** are provided as a Source Data file.

the cell surface expression of P2X7Rs in various CD45[+] cells obtained from WT and Mecp2[308/Y] mice (Supplementary Fig. 2a, b, Supplementary Table 1). In blood, P2X7Rs were expressed in Ly6C[High] monocytes and CD11b[−] cells (Fig. 2a, b). Among leukocytes detected in the cortex, CD206[High]MHCII[Low] macrophages had the most abundant surface P2X7R expression, while Ly6C[High] monocytes and CD206[Low]MHCII[High] macrophages expressed intermediate levels. P2X7R expression in microglia has been documented in previous studies[17,32,33]. Interestingly, among all myeloid populations detected in the cortex, we found that microglia had the lowest levels of surface P2X7Rs (Fig. 2a, c). P2X7Rs were not detected in CD11b[−]CD45[High] cells in the cortex (Fig. 2a, c). Comparing Mecp2[308/Y] to WT mice, the levels of P2X7R expression were similar in blood monocytes (Fig. 2d). In cortex, surface P2X7R expression was comparable in cortical Ly6C[High] monocytes, CD206[Low]MHCII-[High] and CD206[High]MHCII[Low] macrophages (Fig. 2e–g). There was no difference in the levels of microglial P2X7Rs between Mecp2[308/Y] and WT mice (Fig. 2h). These data show that in

both WT and Mecp2[308/Y] mice, surface P2X7R expression is intermediate-to-high in monocytes and macrophages but low in microglia, consistent with recent RNA-seq data[30].

Previous studies have shown that P2X7Rs mediate the increase in membrane permeability to large molecules (≤900 DA) and fluorescent tracers after ATP treatment[34]. We used the organic cation DAPI[2+] (350 DA) to assay uptake as an index of P2X7R-mediated increase in membrane permeability under baseline conditions (i.e., without exogenous ATP) and in the presence of Benzoyl ATP (BzATP), a slowly degradable synthetic form of ATP that preferentially activates P2X7Rs. The baseline DAPI[2+] fluorescence in monocytes, macrophages and microglia was comparable between WT and Mecp2[308/Y] mice, which was similar to the background fluorescence of unstained controls, indicating undetectable baseline DAPI[2+] uptake in viable cells (Supplementary Fig. 2c). After application of 30 μM BzATP for 15 min, DAPI[2+] uptake increased in CD206[High]MHCII[Low] macrophages (P2X7R[High]) derived from both WT and Mecp2[308/Y] mice (Supplementary Fig. 2d). We did not observe BzATP-induced

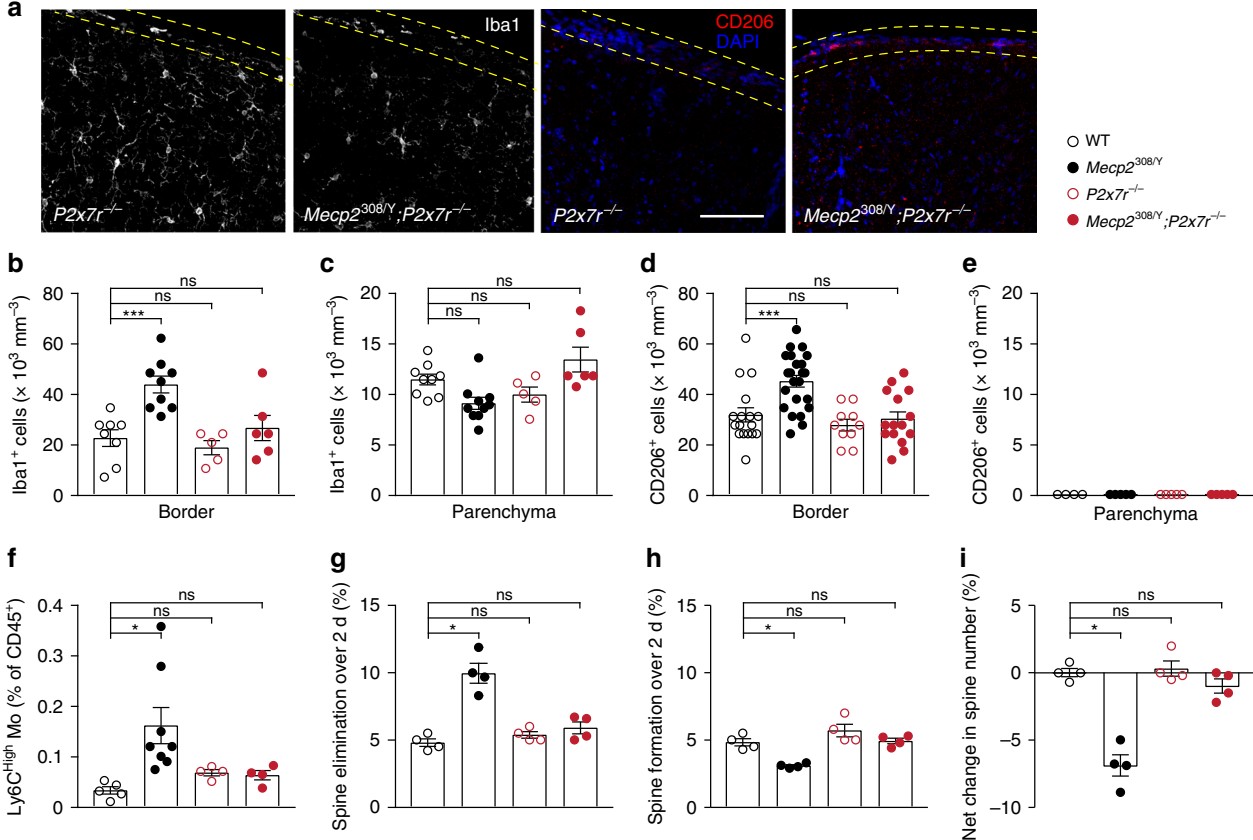

**Fig. 3 P2X7R knockout reduces cortical inflammation and dendritic spine loss in *Mecp2*[308/Y] mice. a** Representative coronal sections of the mouse cortex stained for Iba1 and CD206 in *P2x7r*[−/−] and *Mecp2*[308/Y]; *P2x7r*[−/−] mice (4–6 months old). Scale bar, 80 μm. **b, c** Quantification of Iba1[+] cell density in the cortical border (n = 8, 9, 5, 6; P = 0.0007, 0.9407, 0.9135 vs. WT) and parenchyma (n = 9, 10, 5, 6; P = 0.0732, 0.5915, 0.2845 vs. WT). Black empty circle, WT; black filled circle, *Mecp2*[308/Y]; red empty circle, *P2x7r*[−/−]; red filled circle, *Mecp2*[308/Y]; *P2x7r*[−/−]. **d, e** Quantification of CD206[+] cell density in the cortical border (n = 17, 23, 10, 15; P = 0.0009, 0.8110, 0.9902 vs. WT) and parenchyma (n = 4, 5, 5, 5). **f** Relative percentages of monocytes in the cortex of WT, *Mecp*[308/Y], *P2x7r*[−/−], and *Mecp*[308/Y]; *P2x7r*[−/−] mice (n = 5, 8, 4, 4; P = 0.0137, 0.9034, 0.9418). Monocytes were gated from CD45[High]CD11b[+]MHCII[−]CD11c[−] cells. Relative numbers are expressed as percentages of CD45[+] cells. **g–i** Percentages of dendritic spine elimination (**g**; WT, 4.80 ± 0.29; *Mecp2*[308/Y], 9.98 ± 0.74, P = 0.0286 vs WT; *P2x7r*[−/−], 5.38 ± 0.24, P = 0.2857 vs. WT; *Mecp2*[308/Y]; *P2x7r*[−/−], 5.90 ± 0.44, P = 0.1429 vs. WT), formation (**h**; WT, 4.83 ± 0.27; *Mecp2*[308/Y], 3.08 ± 0.09, P = 0.0286 vs. WT; *P2x7r*[−/−], 5.70 ± 0.47, P = 0.2000 vs. WT; *Mecp2*[308/Y]; *P2x7r*[−/−], 4.93 ± 0.21, P = 0.8857 vs. WT), and net change in total spine number (**i**; WT, 0.03 ± 0.31; *Mecp2*[308/Y], −6.90 ± 0.80, P = 0.0286 vs. WT; *P2x7r*[−/−], 0.33 ± 0.57, P > 0.9999 vs. WT; *Mecp2*[308/Y]; *P2x7r*[−/−], −0.98 ± 0.53, P = 0.2000 vs. WT) over 2 days in 2–3-month-old WT, *Mecp2*[308/Y], *P2x7r*[−/−], and *Mecp2*[308/Y];*P2x7r*[−/−] mice (n = 4 mice per group). Data are presented as mean ± SEM. *P < 0.05, ***P < 0.001, ns not significant, one-way ANOVA followed by Sidak's multiple comparison test was used in **b–d** and **f** and two-tailed Mann–Whitney test was used in (**g–i**). The source data underlying **b–i** are provided as a Source Data file.

DAPI[2+] uptake in microglia (Supplementary Fig. 2d), consistent with their low surface P2X7R expression.

**P2X7R knockout reduces brain inflammation in *Mecp2*[308/Y] mice.** Previous studies have shown that pharmacological blockade of P2X7Rs ameliorates secondary inflammation and damage after spinal cord injury[20]. To test whether ablation of P2X7Rs is effective in reducing cortical inflammation in *Mecp2*[308/Y] mice, we crossed *Mecp2*[308/Y] mice with *P2x7r*[−/−] mice and then backcrossed them to generate *Mecp2*[308/Y];*P2x7r*[−/−] mice. We found that the number of non-microglial Iba1[+] and CD206[+] cells in the cortical border of *Mecp2*[308/Y];*P2x7r*[−/−] mice was not different from that in WT mice (Fig. 3a–e). The relative number of Ly6C[High] monocytes in *Mecp2*[308/Y];*P2x7r*[−/−] mice was comparable to that in WT and *P2x7r*[−/−] mice (Fig. 3f). In *Mecp2*[308/Y] mice, a large fraction of CD206[+] cells express NLRP3 (Supplementary Fig. 3a–c). The NLRP3 inflammasome plays pivotal roles in infection and inflammation and its activation may be mediated by P2X7Rs[15]. Indeed, we observed fewer NLRP3[+] cells in the

cortex of *Mecp2*[308/Y];*P2x7r*[−/−] relative to *Mecp2*[308/Y] mice (Supplementary Fig. 3a–c). Using flow cytometry, we confirmed that surface P2X7Rs were absent in leukocytes in *P2x7r*[−/−] and *Mecp2*[308/Y];*P2x7r*[−/−] mice (Supplementary Fig. 3d). Together, these results indicate that the increase of inflammatory cells in the cortical border of *Mecp2*[308/Y] mice depends on P2X7Rs.

**P2X7R knockout restores cortical dendritic spine stability.** Neuronal dysfunction and impaired synaptic plasticity have been documented in mouse models of RTT[35–37]. In both mice and human patients, cortical neurons with either gain or loss of function mutations in MECP2 have abnormal dendritic branching and reduced spine density[35,38]. We asked whether MECP2 deficiency alters the dynamics of postsynaptic dendritic spines in the mouse cortex, and whether ablating P2X7Rs could reduce cortical spine changes. We crossed *Mecp2*[308/Y] and *P2x7r*[−/−] mice with *Thy1*-YFP transgenic mice that express yellow fluorescent protein (YFP) in layer 5 pyramidal neurons, and examined the dynamics of dendritic spines in the frontal association

cortex, one of the brain regions involved in social behavior[39,40], using transcranial two-photon microscopy[41]. We found that $Mecp2^{308/Y}$ mice with neurological dysfunction and social behavioral deficits exhibited increased dendritic spine elimination (Fig. 3g) and decreased new spine formation (Fig. 3h) as compared to WT mice, leading to a net loss of total spine number over a course of 2 days (Fig. 3i). This reduction in total spine number was not observed in $Mecp2^{308/Y};P2x7r^{-/-}$ mice. There was no difference in the rates of spine elimination and formation between $P2x7r^{-/-}$ mice and WT mice (Fig. 3g–i). These results indicate that P2X7R deficiency restores dendritic spine plasticity in the cortex of $Mecp2^{308/Y}$ mice.

**P2X7R knockout reduces behavior deficits in $Mecp2^{308/Y}$ mice.**
We next investigated whether P2X7R deficiency may improve behavioral outcomes in $Mecp2^{308/Y}$ mice. $Mecp2^{308/Y}$ mice have longer lifespan and milder neurological deficits relative to other mouse models of RTT (e.g., $Mecp2^{tm1.1Bird}$, which are $Mecp2^{-/Y})[25]$. This feature allows us to follow the animals' behavior in male mice up to 12 months. After the postnatal day 45, $Mecp2^{308/Y}$ mice displayed signs of neurological impairments such as the mild deterioration of body condition, altered gait, tremors, and loss of hind limb reflexes (clasping) (Fig. 4a, Supplementary Fig. 4a, b). The majority of $Mecp2^{308/Y}$ mice had normal weights (Supplementary Fig. 4c) and survived for more than 1 year after the onset of neurological dysfunction. Some of these mice may develop severe phenotypes after 8 months of age: marked piloerection, kyphosis and intense tremors causing limited movements as well as seizures. Relative to littermate $Mecp2^{308/Y}$ mice, we found that $Mecp2^{308};P2x7r^{-/-}$ mice had less neurological dysfunction (Fig. 4a, Supplementary Fig. 4a).

$Mecp2^{308/Y}$ mice exhibit abnormal social interactions, which may be relevant to autistic and neuropsychiatric phenotypes of RTT[25,42] and other MECP2 disorders prevalent in males (e.g., MECP2 duplication syndrome)[43]. We examined the animals' social behavior using a 3-chamber social test[44]. Young (2–3 month) $Mecp2^{308}$ mice with behavioral signs of neurological dysfunction were social in that they spent more time interacting with a mouse than with an object (Fig. 4b, c). The sociability index decreased as $Mecp2^{308/Y}$ mice aged (4–6 month) (Fig. 4b, c). Further testing revealed impaired preference for a novel conspecific (social novelty and memory) in both young and adult $Mecp2^{308/Y}$ mice, indicating abnormal social interaction (Fig. 4d, e). Social behaviors in both $Mecp2^{308/Y};P2x7r^{-/-}$ and $P2x7r^{-/-}$ mice were not different from that in WT cohorts (Fig. 4b–e). These data show that P2X7R deficiency corrects social behavioral deficits in $Mecp2^{308/Y}$ mice.

A previous study reported that the locomotor activity of $Mecp2^{308/Y}$ mice decreased slightly in a 30-min open field test, but not over a shorter period of observation[25]. Consistently, in the 3-chamber arena, we did not observe differences in the locomotor activity between WT and $Mecp2^{308/Y}$ mice over a period of 10 min (Supplementary Fig. 4d). When assessed with intensive rotarod training (40 trials), $Mecp2^{308/Y}$ mice showed lower performance relative to WT mice. However, there were no differences in rotarod performances between $P2x7r^{-/-}$, $Mecp2^{308/Y}$, and $Mecp2^{308/Y}; P2x7r^{-/-}$ mice (Supplementary Fig. 4e).

**Leukocyte P2X7Rs mediate social deficits and synapse loss.**
MECP2 transcripts are found in various bone marrow (BM)-derived leukocytes (data source: http://rstats.immgen.org/Skyline/skyline.html). We investigated to what extent MECP2 deficiency outside of CNS, especially in BM-derived immune cells, contributes to the development of neurological dysfunction and

social deficits; and whether P2X7R deficiency is effective in correcting the synaptic and behavioral phenotypes. To address these questions, we transplanted 1-month-old lethally irradiated WT recipients (headshield protected) with BM cell suspensions obtained from WT, $Mecp2^{308/Y}$, $Mecp2^{308/Y};P2x7r^{-/-}$ and $P2x7r^{-/-}$ mice, respectively. These chimeric mice express mutant alleles in peripheral leukocytes including monocytes and monocyte-derived cells but not in yolk-sac derived microglia as well as astrocytes and neurons. We evaluated the animals' neurological scores over the course of 6 months after BM transplantation. The cumulative scores of $Mecp2^{308/Y} \rightarrow$ WT mice were not different from WT mice (Fig. 5a), indicating that MECP2 deficiency in leukocytes is not sufficient to cause neurological dysfunction. Notably, $Mecp2^{308} \rightarrow$ WT chimeras (2–3-month old) showed abnormal social behavior, characterized by reductions in the preference for social novelty, as compared to WT $\rightarrow$ WT mice (Fig. 5b). In contrast, $P2x7r^{-/-} \rightarrow$ WT and $Mecp2^{308/Y};P2x7r^{-/-} \rightarrow$ WT chimera were not different from WT $\rightarrow$ WT mice (Fig. 5b). We did not observe changes in sociability in all these chimeras (Fig. 5c).

Consistent with the alteration of social behavior, the rate of dendritic spine elimination, but not spine formation, was significantly increased in the frontal association cortex of $Mecp2^{308/Y} \rightarrow$ WT chimeras (Fig. 5d, e), leading to a net loss of total spine number (Fig. 5f). There was no difference in the number of dendritic spines eliminated and formed over 2 days between WT $\rightarrow$ WT, $P2x7r^{-/-} \rightarrow$ WT and $Mecp2^{308}; P2x7r^{-/-} \rightarrow$ WT chimeras (Fig. 5d–f). Furthermore, the elimination rates of dendritic spines in $Mecp2^{308/Y} \rightarrow$ WT mice were not different from those in $Mecp2^{308/Y}$ mice (see Fig. 3g). These data indicate that MeCP2 deficiency in BM-derived leukocytes contributes to the abnormalities in social behavior and dendritic spine plasticity through P2X7R-dependent mechanisms.

**Pharmacological block of P2X7Rs improves social behavior.**
Having established the importance of peripheral leukocytes and P2X7Rs in MECP2 deficiency-induced social behavioral deficits, we asked whether pharmacological block of P2X7Rs could improve social behavior in $Mecp2^{308/Y}$ mice. JNJ-47965567 is a blood brain barrier (BBB)-permeable P2X7R antagonist[45]. After systemic administration, JNJ-47965567 targets P2X7Rs in brain and periphery equally due to its high brain to plasma ratio (~1). Consistent with the findings of P2X7R knockout mice, we found that systemic administration of JNJ-47965567 (10 mg/kg/day, i.p.) for 3 days reduced the number of inflammatory myeloid cells in the border of cerebral cortex (Supplementary Fig. 5a–d) and improved the animals' social behavior (Supplementary Fig. 5e, f), indicating the beneficial effects of P2X7R inhibition in $Mecp2^{308/Y}$ mice.

Next, we tested whether the beneficial effects of blocking P2X7Rs in $Mecp2^{308/Y}$ mice can be achieved by primarily inhibiting P2X7Rs in peripheral tissues (e.g., blood, brain's perivascular spaces and meninges). Brilliant Blue G (BBG) is a U.S. Food and Drug Administration-approved drug and has been shown to block P2X7R-mediated increase in membrane permeability and macroscopic currents in cell cultures and spinal cord slices[21,46]. Previous studies have shown that BBG has low penetrance to the CNS parenchyma (brain to plasma ratio ~0.01). After i.p. injection at 50 mg/kg, the concentration of BBG in the brain was estimated to be ~0.002 μM, which is 200 times lower than the $IC_{50}$ of mouse P2X7Rs[47]. Indeed, we found that i.p. injection of 10 mg/kg BBG colored plasma samples, abdominal tissues and highly vascularized dura mater, but did not stain the cerebral cortex (excluding meninges) (Supplementary Fig. 6a, b). Following 3–5-day treatment, BBG was detected in plasma

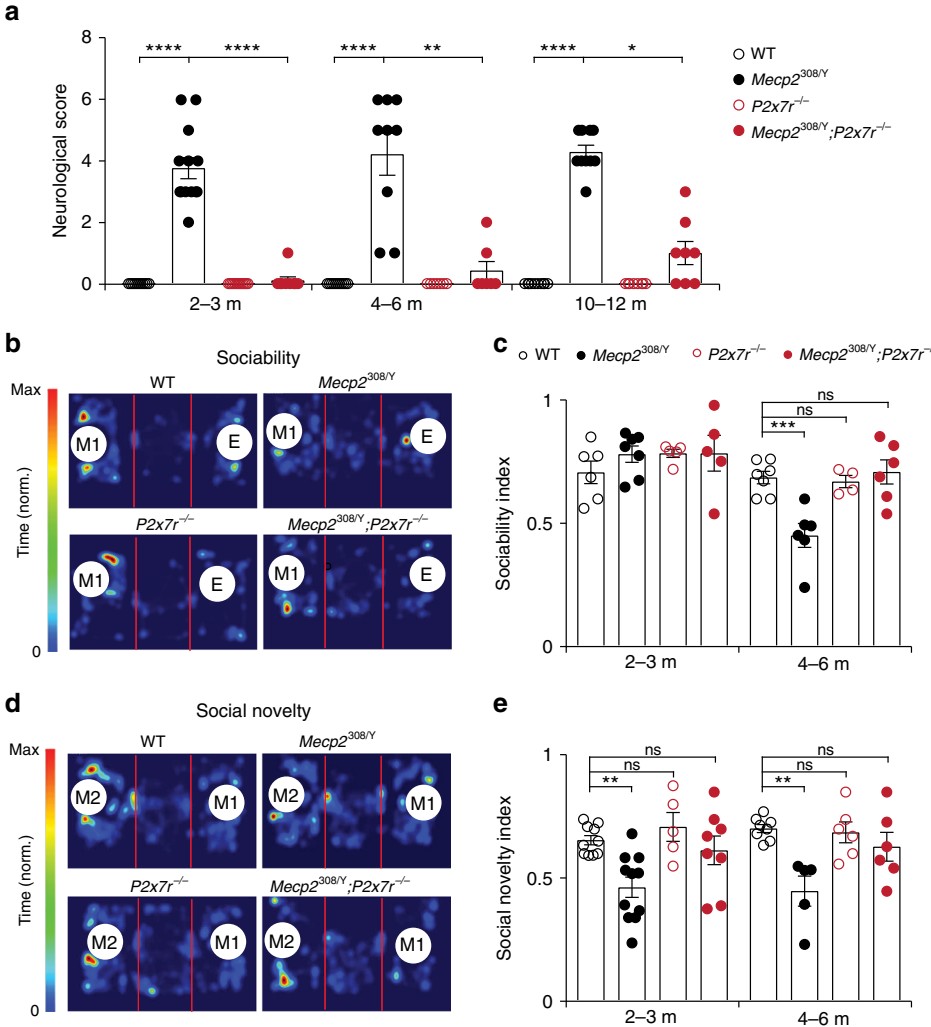

**Fig. 4 P2X7R knockout ameliorates behavioral deficits in _Mecp2_[308/Y] mice. a** Assessments of neurological function in WT, _Mecp2_[308/Y], _P2x7r_[−/−] and _Mecp2_[308/Y]; _P2x7r_[−/−] mice at ages of 2–3 ($n$ = 11, 13, 13, 9 mice; _Mecp2_[308/Y] vs. WT, $P < 0.0001$; _Mecp2_[308/Y];_P2x7r_[−/−] vs. _Mecp2_[308/Y], $P < 0.0001$), 4–6 ($n$ = 11, 9, 7, 7 mice; _Mecp2_[308/Y] vs. WT, $P < 0.0001$; _Mecp2_[308/Y];_P2x7r_[−/−] vs. _Mecp2_[308/Y], $P = 0.0041$), and 10–12 months ($n$ = 10, 10, 8, 8 mice; _Mecp2_[308/Y] vs. WT, $P < 0.0001$; _Mecp2_[308/Y];_P2x7r_[−/−] vs. _Mecp2_[308/Y], $P = 0.0240$). Scores 0, 1, and 2 indicate absence, mild and severe manifestation of neurological dysfunction, respectively. **b** Representative heat-maps illustrating time that mice (4-month old) spent at each location. E, empty cage, M1, Mouse 1 (first stranger). **c** Sociability index measured in WT, _Mecp2_[308/Y], _P2x7r_[−/−], and _Mecp2_[308/Y];_P2x7r_[−/−] at 2–3 ($n$ = 6, 7, 5, 5 mice; $P = 0.4938$, 0.5289, 0.5219 vs. WT) and 4–6 months ($n$ = 7, 6, 4, 6 mice; $P = 0.0006$, 0.9917, 0.9718 vs. WT) of age. **d** Representative heat-maps showing time that mice (4-month old) spent at each location. M2, Mouse 2 (second stranger). **e** Social preference index measured in WT, _Mecp2_[308/Y], _P2x7r_[−/−], and _Mecp2_[308/Y];_P2x7r_[−/−] mice at 2–3 ($n$ = 10, 11, 5, 8 mice; $P = 0.0016$, 0.7954, 0.8475 vs. WT) and 4–6 ($n$ = 8, 5, 6, 6 mice; $P = 0.0013$, 0.9926, 0.5844 vs. WT) months of age. Data are presented as mean ± SEM. *$P < 0.05$, **$P < 0.01$, ***$P < 0.001$, ****$P < 0.0001$, ns not significant, Kruskal–Wallis test followed by Dunn's multiple comparison test was used in (**a**), two-way ANOVA followed by Sidak's multiple comparison test was used in (**c**, **e**). The source data underlying **a**, **c**, and **e** are provided as a Source Data file.

(~30 µM) of both WT and _Mecp2_[308/Y] mice (Supplementary Fig. 6c), using a method with a sensitivity of 0.01 µM. BBG was also detected in both dura (~5 µM) and pia mater (~12 µM). BBG was not detected in the cortical parenchyma >30 µm away from the pial surface (Supplementary Fig. 6c). We, therefore, reasoned that i.p. administration of BBG would primarily inhibit P2X7Rs in circulating immune cells and macrophages associated with meninges and cortical border. Notably, we found that BBG treatment for 3–5 days significantly reduced the relative numbers of Ly6C[High] monocytes and CD206[High]MHCII[Low] and CD206[Low]MHCII[High] macrophages in the cortex of _Mecp2_[308/Y] mice, with no significant effects on the microglial fraction (Fig. 6a–d).

Consistent with the results in chimeric mice, we found that systemic administration of BBG had no effects on the neurological function of _Mecp2_[308/Y] mice (Fig. 6e). At 2–3 months of age, there was no difference in sociability between BBG-treated _Mecp2_[308/Y] cohorts and BBG-treated WT mice (Fig. 6f). At the same age, BBG treatment substantially improved the preference for social novelty in _Mecp2_[308/Y] mice (Fig. 6g). In separate experiments, we identified _Mecp2_[308/Y] mice with low preference for social novelty (index < 0.55) and treated them with BBG (3 days) until we observed an improvement in their social behavior. Following the suspension of treatment for 3 days, this improvement in social behavior in _Mecp2_[308/Y] mice diminished by day 6 (Fig. 6h). These results further support that MECP2 deficiency in peripheral leukocytes contributes to social behavioral deficits in _Mecp2_[308/Y] mice through P2X7R-dependent mechanisms.

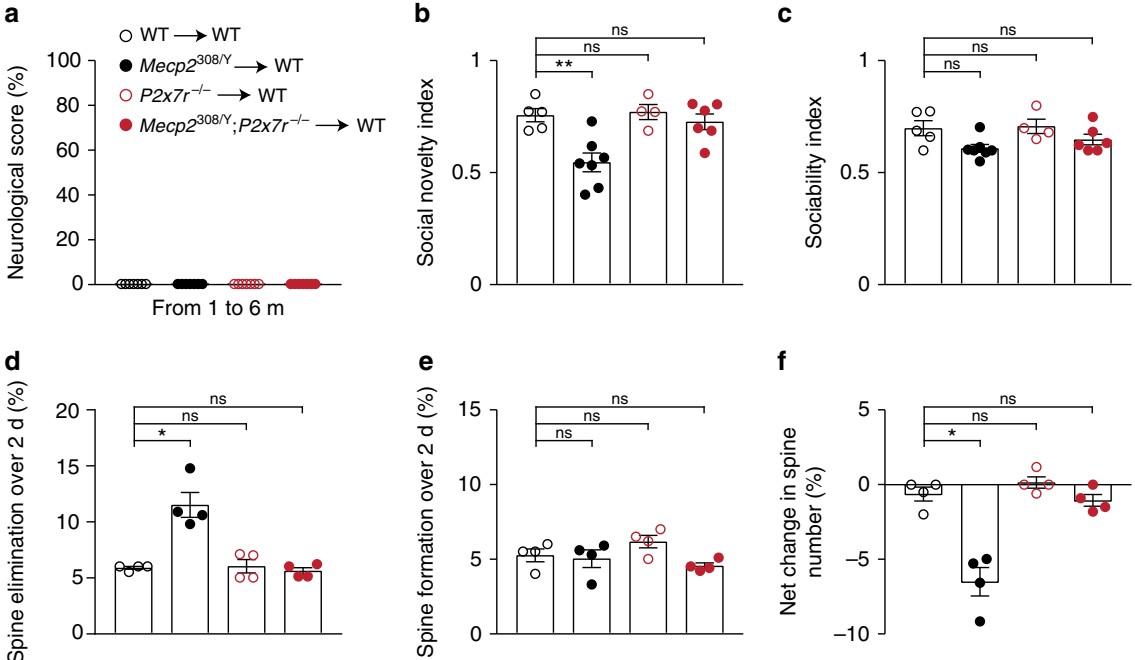

**Fig. 5 Leukocyte P2X7Rs mediate social behavior deficits and cortical spine loss. a** Neurological scores measured in WT, *Mecp2*[308/Y], *P2x7r*[−/−] and *Mecp2*[308/Y];*P2x7r*[−/−] chimeric mice (*n* = 7 mice per group). **b, c** Social novelty (**b**; *n* = 5, 7, 4, 6 mice; *P* = 0.0029, 0.9988, 0.9718) and sociability (**c**; *n* = 5, 7, 4, 6 mice; *P* = 0.0766, 0.9993, 0.5508) index measured in various chimeric mice. **d–f** Quantification of spine elimination (**d**; WT, 5.88 ± 0.13; *Mecp2*[308/Y], 11.53 ± 1.12, *P* = 0.0286 vs. WT; *P2x7r*[−/−], 6.03 ± 0.59, *P* > 0.9999 vs. WT; *Mecp2*[308/Y]; *P2x7r*[−/−], 5.60 ± 0.29, *P* = 0.7714 vs. WT), spine formation (**e**; WT, 5.25 ± 0.43; *Mecp2*[308/Y], 5.03 ± 0.59, *P* = 0.8571 vs. WT; *P2x7r*[−/−], 6.18 ± 0.43, *P* = 0.1714 vs. WT; *Mecp2*[308/Y]; *P2x7r*[−/−], 4.55 ± 0.19, *P* = 0.2857 vs. WT) and changes in total spine number (**f**; WT, −0.63 ± 0.47; *Mecp2*[308/Y], −6.53 ± 0.96, *P* = 0.0286 vs. WT; *P2x7r*[−/−], 0.15 ± 0.38, *P* = 0.4857 vs. WT; *Mecp2*[308/Y]; *P2x7r*[−/−], −1.05 ± 0.40, *P* = 0.6571 vs. WT) in chimeric mice (*n* = 4 mice per group). Data are presented as mean ± SEM. *\*P* < 0.05, ns not significant, one-way ANOVA followed by Sidak's multiple comparison test was used in (**b, c**) and two-tailed Mann–Whitney test was used in (**d–f**). The source data underlying **a–f** are provided as a Source Data file.

## Discussion

In this study, we investigated the pathophysiological roles of P2X7Rs in a mouse model of RTT with MECP2 deficiency. We found that *Mecp2*[308/Y] mice had an increased number of inflammatory cells expressing P2X7Rs in the border of the cerebral cortex. Depletion of P2X7Rs reduced the number of inflammatory cells in the cortical border, restored cortical dendritic spine plasticity, and improved the animals' neurological function and social behavior. Using bone marrow chimera and pharmacology to inhibit P2X7Rs in peripheral immune cells, we further showed that MECP2 deficiency in leukocytes contributed to social behavior defects via P2X7R-dependent mechanisms. Together, our data reveal a pivotal role of P2X7Rs and non-microglial myeloid cells in dendritic spine pathology and social behavioral deficits in RTT.

In *Mecp2*[308/Y] mice, we found that P2X7Rs are most abundantly expressed on the surface of a subset of resident macrophages in the cortical border. Yolk-sac derived microglia and macrophages and monocyte-derived macrophages are cells of myeloid lineage of different origins, located in different compartments of CNS (parenchymal and nonparenchymal)[31]. Although much of literature in the field of neuroscience does not distinguish myeloid cell types in the brain, increasing evidence indicate that microglia and nonparenchymal macrophages have different molecular signatures and function[48]. In this study, we defined microglia in situ as ramified Iba1+CD206− cells, and ex vivo using a combination of cell surface markers, such as CD11b, CD45, CD64, CD206, MHCII, and CD38, for flow cytometry staining. Unlike in neurodegeneration and injury[49,50], we did not observe morphological signs of microglial activation in *Mecp2*[308/Y] mice and these *Mecp2*[308/Y] microglia displayed

immune quiescent molecular signatures (P2X7R[Low], CD11c[Low], and MHCII[Low]). The role of microglia in the pathogenesis of RTT has been controversial due to the lack of genetic tools to selectively manipulate MECP2 in microglia without affecting other myeloid populations[51,52]. A recent study showed that long-lived myeloid cells in the brain, including microglia, do not contribute to RTT-like neurological abnormalities in *Mecp2*[tm1.1Bird] mice, but may contribute to neural circuit dysfunction independently of MECP2 deficiency[53]. By gating out monocytes, neutrophils, and lymphocytes, we analyzed the surface expression of P2X7Rs in microglia and macrophages, separately. We found that the surface expression of P2X7Rs was robust in CD206[High]MHCII[Low] macrophages, but very low in microglia, in both WT and *Mecp2*[308/Y] mice (other myeloid subsets expressed intermediate levels). This finding seems to be at odds with previous reports of high P2X7R expression in microglia in cognitive-related pathologies[32]. Such discrepancy can be attributed to minimal microglia activation in the cortex of *Mecp2*[308] mice, as well as our excluding macrophages in the flow cytometry analysis. In this regard, a recent study describing a single-cell atlas of mouse brain myeloid cells showed that P2X7R expression is high in leptomeningeal macrophages, but very low in microglia[30].

Our study underscores the role of P2X7Rs and nonmicroglial myeloid cells in mediating synaptic and social behavioral deficits in *Mecp2*[308/Y] mice. This was demonstrated by a series of genetic and pharmacological experiments. First, P2X7R knockout reduced inflammation and dendritic spine loss in the cortex of *Mecp2*[308/Y] mice, as well as the animal's behavioral defects. Second, using *Mecp2*[308] → WT chimeric mice, we showed that MECP2 deficiency in BM-derived leukocytes was sufficient to alter dendritic spine dynamics and social behavior, whereas

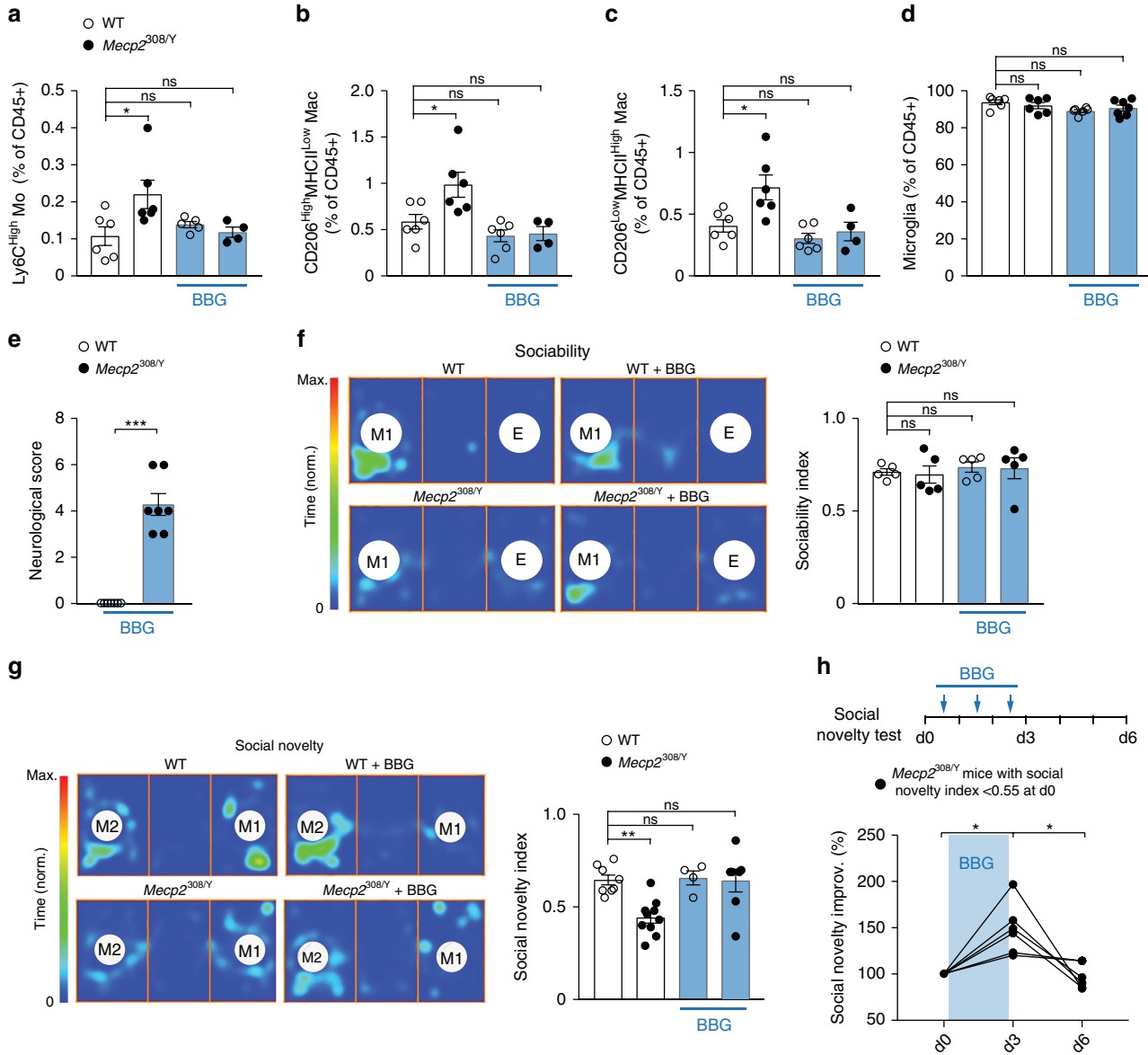

**Fig. 6 Pharmacological inhibition of P2X7Rs reduces inflammation and improves social behavior. a** Percentages of Ly6C<sup>High</sup> monocytes in WT and *Mecp2*<sup>308/Y</sup> mice with or without BBG treatment (10 mg/kg/day for 5 days; *n* = 6, 6, 5, 4 mice; *P* = 0.0188, 0.8078, 0.9912). Ly6C<sup>High</sup> monocytes were gated from CD45<sup>High</sup>CD11b<sup>+</sup>MHCII<sup>−</sup>Ly6G<sup>−</sup> cells. **b**–**d** Percentages of CD206<sup>High</sup>MHCII<sup>Low</sup> (**b**; *n* = 6, 6, 6, 4 mice; *P* = 0.0202, 0.5922, 0.7773) and CD206<sup>Low</sup>MHCII<sup>High</sup> (**c**; *n* = 6, 6, 6, 4 mice; *P* = 0.0135, 0.6667, 0.9661) macrophages and microglia (**d**; *n* = 6, 6, 6, 7 mice; *P* = 0.7796, 0.0869, 0.3334) in WT and *Mecp2*<sup>308/Y</sup> mice with or without BBG treatment (10 mg/kg/day for 5 days). **e** Neurological scores of WT and *Mecp2*<sup>308/Y</sup> mice treated with BBG (*n* = 7 mice per group; *P* = 0.0006). **f** Sociability test (*n* = 5 mice per group). Left: representative heat-maps showing time spent at each location. M1, Mouse 1; E, empty cage. Right: Sociability index measured in WT and *Mecp2*<sup>308/Y</sup> mice with or without BBG treatment (*P* = 0.9931, 0.9591, 0.9805). **g** Social novelty test. Left: representative heat-maps showing time spent at each location. M1, Mouse 1 (1st stranger); M2, Mouse 2 (2nd stranger). Right: Social novelty index measured in WT and *Mecp2*<sup>308/Y</sup> mice with or without BBG treatment (*n* = 8, 10, 4, 7 mice; *P* = 0.0018, 0.9977, 0.9999). **h** Social novelty test (*n* = 6 mice per group). Top: experimental timeline. *Mecp2*<sup>308/Y</sup> mice with social novelty index <0.55 at day 0 were given BBG treatment (10 mg/kg/day) for 3 days. Social novelty was reassessed on day 3 and 6. Bottom: Quantification of behavior improvement in the social novelty test (*P* = 0.0313, 0.0313). Data are presented as mean ± SEM. *\*P* < 0.05, *\*\*P* < 0.01, *\*\*\*P* < 0.001, ns, not significant, two-way ANOVA followed by Sidak's multiple comparison test in **a**–**d**, **f** and **g**, two-tailed Mann–Whitney test in (**e**) and two-tailed Wilcoxon test in (**h**). The source data underlying **a**–**h** are provided as a Source Data file.

depletion of P2X7Rs in BM-derived leukocytes prevented these changes. When generating BM chimeras, we irradiated mice with head-shield protection to prevent artifactual and confounding effects of BM-derived cells infiltrating into the CNS parenchyma[54]. Third, systemic administration P2X7R antagonists, either a BBB-permeable compound (JNJ-47965567) or a compound with low penetrance of brain parenchyma (BBG), reduced the number of inflammatory cells in the cortical border and

reversed social behavioral deficits in *Mecp2*<sup>308/Y</sup> mice. Following i.p. injection, BBG mainly distributed in peripheral tissues (e.g., blood and meninges). Although detected in the border (within 30 μm of pial surface) of the cortex, BBG was not detected in the deep cortex (30 μm away from pial surface), suggesting its minimal impact on parenchymal cells such as microglia. We cannot rule out the possibility that BBG may also affect P2X7Rs expressed on endothelial cells, pericytes and astrocyte processes

forming part of BBB and cortical border. Together, these data indicate that peripheral leukocytes are involved in synapse pathology and social behavioral deficits in RTT through mechanisms dependent on P2X7Rs.

CD206$^{High}$MHCII$^{Low}$ macrophages cooperate with CD206$^{Low}$ MHCII$^{High}$ macrophages to amplify the pathological effects in $Mecp2^{308/Y}$ mice. Although the majority of CD206$^{High}$MHCII$^{Low}$ macrophages are long-lived and derived from yolk-sac progenitors, a small fraction of them are replenished through hematopoietic progenitors[30]. Whether in the context of RTT a higher fraction of these macrophages are replenished through circulating monocytes remains to be determined. Future studies using in vivo depletion of specific macrophage subsets, adoptive cell transfer and $Mecp2^{308/Y};P2x7r^{-/-} \rightarrow Mecp2^{308/Y}$ chimeras may help address these questions.

At least four splice variants have been described for the mouse $P2rx7$ gene[15]. P2X7a is the full-length variant forming the functional P2X7Rs in macrophages, whereas P2X7k is generated by alternative splicing and forms functional channels in naïve and regulatory T cells[55]. Other variants such as P2X713b and P2X713c are truncated and do not form functional channels[56]. The $P2x7r^{-/-}$ mice (Pfizer) used in this study lack of P2X7a and P2X7k, in contrast to other knockout mice (Glaxo) that are P2X7a$^-$ P2X7k$^+$ [57]. Moreover, all $P2x7r^{-/-}$ mice may still express truncated splice variants that could regulate the function of cell surface receptors[58]. Given the differential effects on the animals' neurological function from genetic knockout and pharmacological block of P2X7R channels, it remains to be determined whether the splice variants forming truncated receptors may have a role in preventing MeCP2 deficiency-induced neurological dysfunction.

The molecular mechanisms underlying the pathophysiological roles of P2X7Rs in RTT remain unclear. Targeting a human neuronal ion transporter to correct chloride imbalance and synaptic dysfunction has proven to be beneficial for RTT[37]. Interestingly, the human P2X7Rs spend less time in the close state after substitution of $[Cl^-]_e$ by glutamate[59], an experimental condition that may be relevant to RTT where high levels of CSF glutamate, reduced number of excitatory synapses and impaired synaptic scaling have been documented[36,60]. In peripheral macrophages, P2X7R (P2X7a) channel opening by ATP causes Na$^+$ and Ca$^{2+}$ influx and K$^+$ efflux, promoting the maturation and release of the pro-inflammatory cytokine IL-1β through NLRP3 inflammasome dependent mechanisms, leading to pyroptosis[15]. We observed a higher number of NLRP3$^+$CD206$^+$ macrophages in the cortical border of $Mecp2^{308/Y}$ mice as compared to $Mecp2^{308/Y};P2x7r^{-/-}$ mice, as well as $Mecp2^{308/Y}$ mice treated with P2X7R blockers, suggesting a fraction of these CD206$^+$ macrophages may replenish through blood monocytes or local macrophage progenitors. In the CNS, IL-1 signaling has been associated with neuronal dysfunction[61]. P2X7R-mediated IL-1β release may potentiate other inflammatory pathways. For example, peripherally generated TNFα has been shown to be involved in cortical synaptic remodeling[41,62] and is released upon activation of P2X7Rs[63]. Meningeal inflammation may prime IFNγ-mediated immune responses which have been associated with interneuron dysfunction and social deficits[64]. Other possible pro-inflammatory pathways downstream of P2X7R activation include calcium-dependent PGE2 release, which has been implicated in sickness behavior[65]. In an Alzheimer's disease mouse model, P2X7Rs mediate beta-amyloid-induced chemokine release and lack of P2X7Rs reduces beta-amyloid deposition and improves synaptic plasticity and memory[66]. Furthermore, activation of P2X7Rs leads to the opening of accessory channels such as Pannexin1 (hemi)channels, which mediates ATP release and supports P2X7R activation within an inflammatory microenvironment[21]. The cooperation between P2X7Rs and Pannexin1 (hemi)channels and its implications in physiology and pathology have been extensively studied in astrocytes[67,68], but it is still unclear whether, and if so how, this cooperation exists in monocytes and macrophages in the CNS. Nevertheless, pro-inflammatory pathways triggered by P2X7Rs, such as inflammasome activation and IL-1β release, may also occur independently of Pannexin1[69].

In summary, our study reveals that P2X7R inhibition reduces cortical inflammation, abnormal dendritic spine remodeling and various behavioral deficits of $Mecp2^{308/Y}$ mice. Therefore, P2X7Rs may serve as potential therapeutic targets in the treatment of RTT and other neurodevelopmental disorders.

## Methods

**Transgenic mice**. All mice used in this study were obtained from Jackson Laboratory. *Thy1*-YFP-H mice (stock 003782) expressing YFP in layer 5 pyramidal neurons and *Mecp2*[308] mice (stock 005439) were crossed to C57BL/6J mice (stock 000664) for 12 generations by provider. *P2x7r*[−/−] mice (stock 005576) were backcrossed to C57BL/6J mice for 7 generations. For dendritic spine imaging, *Thy1*-YFP mice were crossed with *Mecp2*[308] mice and *P2x7r*[−/−] mice. One to 12-month old male mice were used for all the experiments. Littermate mice were group-housed in temperature-controlled rooms on a 12-h light–dark cycle and were randomly assigned to different treatment groups. The group size was determined based on previous studies using the same methodologies[41]. All mice were generated and maintained at the New York University Skirball Institute or Columbia University specific pathogen-free animal facility. All animal procedures in this study were approved by the Institutional Animal Care and Use Committee (IACUC) of New York University and Columbia University as consistent with National Institutes of Health (NIH) Guidelines for the Care and Use of Laboratory Animals.

**Generation of bone marrow chimeric mice**. One-month-old recipient mice were exposed to 1200 rad of whole-body gamma irradiation in 2 sessions, 3 h apart. Head protection was used in all cohorts of recipient mice during irradiation to minimize potential artifacts caused by cranial irradiation. Donor BM cells were extracted from the femur of donor mice and passed through a 70-μm cell strainer. Recipient mice were injected with donor BM cells ($3–5 \times 10^6$ cells) through the retro-orbital venous sinus 1–3 h after the second session of irradiation. After transplantation, the animals' body weight was monitored every other day. For the first 2 weeks after irradiation, animals' drinking water was supplemented with 2 mg/ml sulfamethoxazole and 0.4 mg/ml trimethoprim. One to 6 months after BM reconstitution, all BM chimeric mice showed no signs of neurological dysfunction. The presence of *Mecp2*[tm1Hzo] and *P2x7r*[tm1Gab] alleles was confirmed by PCR at least 4 weeks after BM transplantation.

**Immunohistochemistry and data analysis**. Mice were deeply anesthetized and perfused with 20 ml Ca$^{2+}$/Mg$^{2+}$-free Dulbecco's phosphate-buffered saline (DPBS) supplemented with 5 mM EDTA. Brain was rapidly removed and fixed overnight in 4% paraformaldehyde at 4 °C. Tissue was rinsed 3 times with PBS and sectioned at 200 μm with a Leica vibratome (VT 1000S). Sections were permeabilized in 1% Triton X-100 in PBS for 24 h and blocked with a solution containing 0.1% Triton X-100 and 5% normal goat serum for 1 h. Sections were incubated overnight with primary antibodies: rabbit anti-Iba1 (Wako, 019-19741) and Alexa fluor 647 anti-mouse CD206 (BioLegend, 141712), goat anti-NLRP3 (Abcam, ab4207 or ab214185). Sections were then washed 3 times with PBS/0.05% Tween-20 and incubated for 1 h with secondary antibodies: Alexa Fluor 594-conjugated goat anti-rabbit IgG (Abcam, ab150088) or Alexa Fluor 555-conjugated donkey anti-goat IgG (Abcam, ab150130). Sections were washed as before, incubated with DAPI (1 ng/ml) for 10 min, washed, and mounted in MoWiol 4–88/glycerol for imaging. Images were obtained using a Zeiss 700 confocal microscope.

The density of NLRP3$^+$, CD206$^+$, and Iba1$^+$ cells in the border and parenchyma region of the cortex was quantified from coronal brain sections. The number of cells per 320 × 30 × 30 μm border tissue or 320 × 290 × 30 μm parenchyma was counted, and the cell density was calculated. To assess the morphological activation of microglia, the area of Iba1$^+$ cells was calculated using ImageJ software[21]. Briefly, fluorescence images (150 × 150 μm) of the cortical parenchyma (30 μm below pial surface) were collected with confocal microscopy. The binary image was created by setting a threshold of $1.2 \times F_{background}$. $F_{background}$ was determined by measuring the average fluorescence in a region without Iba1$^+$ cells. The area occupied by the microglial somata and processes was expressed as a percentage of the total area examined.

**Tissue preparation for flow cytometry**. Blood was collected from the retro-orbital vein in anesthetized mice before perfusion. Mice were perfused with 25 ml of Ca$^{2+}$/Mg$^{2+}$-free DPBS (Sigma, D8537) supplemented with 5 mM EDTA.

Cortex, subcortex and cerebellum were removed and placed in DPBS supplemented with HEPES (10 mM) and 2.5% FCS. Single cell suspensions of blood, cortex, subcortex, and cerebellum were prepared. The brain and cerebellum samples were deprived of dura mater, but not leptomeninges. Dura mater-deprived brain and cerebellum samples were minced with scissors and incubated with 150 U of collagenase D (Roche, 11088858001) at 37 °C for 30 min. Collagenase was inactivated by adding 12.5 mM EDTA for an additional 5-min incubation at 37ºC. Digested material was mechanically dissociated and passed through a 70-μm cell strainer, followed by a centrifugation at 500g, at 4 °C, for 30 min in a continuous 38% Percoll gradient. Cell pellets were resuspended in FACS buffer ($Ca^{2+}/Mg^{2+}$-free PBS containing 0.5% bovine serum albumin (BSA) and 1 mM EDTA). For DAPI uptake experiments, cells were incubated with 0.2 μM DAPI alone or together with 30 μM BzATP (TOCRIS, 3312) for 15 min, at room temperature. Afterwards, DAPI and BzATP were removed by washing cells twice in FACS buffer. Nonspecific binding to FC receptors was blocked by incubation with anti-CD16/32 antibody (BioXcell, BE0307) for 15 min at 4 °C. Cells were washed and stained with antibody panel for cell surface markers expressed by lymphocytes and myeloid cells. The antibodies are listed in Supplementary Table 3. Flow cytometry was performed on a LSRII (BD) and analyzed with FlowJo V 10 (Treestar).

**Flow cytometry analysis**. Cells were first separated by their physical properties, i.e., size and complexity were analyzed by measuring the intensity of light scattered along the path of the laser (FSC) and the light scattered at 90° angle relative to the laser beam (SSC). Cells falling within a range of 40–250 K in the FSC and 0–200 K in the SSC were gated for analysis. Blood and brain leukocytes were included within this range of $FSC^{Low}$ (<200 K) and $SCC^{Low}$ (<150 K) cells. Only singlets and cells taking up low amounts of DAPI ($DAPI^{Low}$) were used for further multiparametric analysis of cell surface marker expression and DAPI uptake experiments. A total of 150–200 K events per sample were acquired on an LSRII (BD) and analyzed with FlowJo V 10.

**Transcranial two-photon microscopy and imaging analysis**. Dendritic spines in the frontal association cortex (+2.8 mm from Bregma and +1 mm from midline) were imaged through a thinned-skull window using a two-photon microscope[41,70]. Briefly, in anesthetized mice (100 mg/kg ketamine and 15 mg/kg xylazine, i.p.), a midline scalp incision exposed the skull. The animal's head was immobilized, and a cranial window was created by thinning a circular area (~200 μm in diameter) of the skull to approximately 20 μm in thickness by using a high-speed drill. Upon completion of the skull thinning, the anesthetized mouse was placed under a two-photon microscope. Image stacks of dendritic segments within a depth of 100 μm from the pial surface were obtained with two-photon laser tuned to 920 nm and using a 1.1 NA 60× objective immersed in artificial cerebrospinal fluid. High-magnification (66.7 × 66.7 μm; 512 × 512 pixels; 0.75 μm step) imaging was used to obtain high-quality images for dendritic spine analysis. After completing the imaging session, the scalp was sutured with 6-0 silk and the animal was returned to the home cage until the next viewing.

Analysis of dendritic spine plasticity was performed using the NIH ImageJ software[41]. Briefly, spines were considered stable if they were present in both views, eliminated if they were present in the 1st view but not in the second view or newly formed if they were present in the second view but not in the first view. The elimination and formation rates were measured as the number of spines eliminated or formed in the 2nd view divided by the number of spines existing in the first view.

**Neurological score**. Neurological score was calculated as the sum of scores from below behavioral tests (0, null; 8, lethal neurological dysfunction). The symptomatic $Mecp2^{308/Y}$ mice were scored as high as ~6.

General body condition: Mice were picked up at the base of the tail and tested by an experienced researcher by passing fingers through back and pubic bones in order to note muscle tone and whether bones were palpable but not prominent. Other obvious health problems, such as rectal prolapse, coat appearance (dermatitis, wounds, and piloerection), malocclusion, and hunched posture, were evaluated. Score: 0, indicates clear coat and eyes and normal stance and breathing pattern; 1, mild changes in coat appearance (ungroomed or dull coat) and/or mild hunched posture, normal breathing; 2, piloerection, narrowed eyes, hunched posture, reduced muscle tone, and/or obvious tremors and difficulty for breathing normally.

Gait: Mice were removed from home cages and placed on a flat surface with heads facing away from the observer. Mice presenting a normal walking were assigned a score of 0. Normal walking is defined as body weight supported on all limbs, abdomen not touching the ground, both front and hind limbs walking evenly and absence of tremors. Mice showing limp or tremors during walking were assigned a score of 1. If the mouse presented difficulty in moving forward, dragged its abdomen along the ground or showed severe tremors during walking, it received a score of 2. In a separate testing, gait alterations were measured using a foot-print analysis.

Hindlimb clasping: Mice were suspended from the tail with the head pointing downward. The hindlimb position was observed for 5 s. If hindlimbs consistently splayed outward, away from the abdomen, it was assigned a score of 0. If one or

two hindlimbs retracted toward the abdomen for more than 50% of the time suspended, it received a score of 1. If both hindlimbs entirely retracted and touched the abdomen for more than 50% of the time suspended, it received a score of 2.

Tremors: Mice were maintained on the hand of an experienced researcher and the absence of palpable tremors (score 0), intermittent tremors (score 1), and severe intermittent or permanent tremors (score 2) was recorded.

**Social behavior**. The three-chamber test was used for evaluating sociability and preference for social novelty in mice[44]. In brief, animals were habituated for 30–45 min the day before the test and 10 min on the test day (pre-test). During the habituation the mice were allowed to freely explore a 60 × 40 × 23 cm plexiglass arena divided into three equally sized, interconnected chambers (left, center, and right). Sociability was measured during the second 10-min period in which the subject could interact either with an empty wire cage (object condition) or a wire cage containing an age and sex-matched stranger conspecific (social condition, mouse 1). Sociability was defined as the time spent interacting (sniffing, crawling upon) with the cage containing a mouse (mouse 1) divided by the sum of the time interacting with occupied and empty cages. The preference for social novelty was assayed by introducing a second stranger mouse (novelty condition, mouse 2) into the wire cage that was previously left empty. Preference for social novelty was calculated as the time spent interacting with the mouse 2 divided by the sum of time interacting with mouse 1 and mouse 2. Time spent interacting with either empty cage or cages containing mouse 1 and/or mouse 2 was recorded using the automated Any-Maze 6.1 software and confirmed by visual inspection of video tracks. Representative heatmaps were generated using either EthoVision 11.5 or Any-Maze software.

Social behavior analysis was performed using EthoVision 11.5 or Any-Maze software. For each experiment, the background level was set at the minimum intensity output obtained using graphical pseudo-color maps. The sociability index was calculated as $T_{M1}/(T_{M1} + T_E)$ where $T_{M1}$ was the time that the test mouse spent interacting with a stranger mouse 1 (M1) and $T_E$ was the time that the test mouse spent interacting with the object. Preference for social novelty was calculated as $T_{M2}/(T_{M1} + T_{M2})$ where $T_{M2}$ was the time interacting with a novel mouse 2 (M2) and $T_{M1}$ was the time interacting with mouse 1.

**Rotarod**. An EZRod (rotarod) system was used to test motor coordination and motor learning. The dimension of the test chamber was 44.5 × 14 × 51 cm. Animals were placed on the motorized rod (30 mm in diameter) in the chamber. The rod remained stationary for 30 s, and then the rotation speed gradually increased from 0 to 80 R.P.M. over the course of 2 min. The latency to fall and rotation speed were recorded when the animal was unable to keep up with the increasing speed. Rotarod training was performed in a 40-trial training session. Performance was measured as the average speed (in R.P.M.) that the animals achieved in the training session.

**Drug injection**. BBG (SIGMA, 27815) was dissolved in DPBS (1 mg/ml) and injected i.p. at a dosage of 10 mg/kg/day for 3–5 days. JNJ-47965567 (SIGMA, SML1708) was first dissolved in 100% DMSO (25 mg/ml) and then diluted 100 times in DPBS and injected i.p. twice a day, at a dosage of 10 mg/kg/day for 3 days.

**Quantification of BBG in plasma and brain tissues**. Both plasma and brain tissues were collected for BBG measurement. To extract plasma, blood was collected using heparinized capillary tubes from the retro-orbital vein of the anesthetized mice. Blood samples were centrifuged at 1400g for 10 min at 4 °C and clear plasma supernatants were collected and stored at −20 °C for 1–2 days. To collect cortical tissues, mice were perfused with 25 ml of $Ca^{2+}/Mg^{2+}$-free DPBS supplemented with 5 mM EDTA and quickly decapitated. The dura mater was dissected from the internal side of skull in ice-cold DPBS containing 0.5% BSA and 1 mM EDTA. The cortex was sectioned in ice-cold DPBS containing 0.5% BSA and 1 mM EDTA using a Leica Vibratome (VT1000S; 0.1 mm/s, 70 Hz). Sections of the superficial layer of the cortex (30 μm in thickness, containing the pia mater) were collected and homogenized. The remaining cortical parenchyma was sectioned at 300 μm in thickness. Brain tissues and meninges were mechanically dissociated in DPBS supplemented with 0.5% BSA and 1 mM EDTA and stored at −20 °C for subsequent BBG analysis. The BBG concentration was measured using a spectrophotometer at the maximal absorbance for BBG (576 nm at pH = 7.2–7.4) against a standard curve which was generated by adding the known amount of BBG (0.005–100 μM) to the untreated brain and blood samples. Using this method, we were able to detect BBG concentration at a range of 0.01–100 μM.

**Statistics**. Prism software (GraphPad 8.0, La Jolla, CA) was used to conduct the statistical analysis. Summary data were presented as mean ± SEM. Tests for differences between two populations were performed using unpaired two-tailed $t$ or Mann–Whitney test, or paired two-tailed Wilcoxon tests. Multiple group comparisons were performed with one or two-way ANOVA followed by Sidak's multiple comparison test, or Kruskal–Wallis test followed by Dunn's multiple comparison test as indicated in figure legends. No data points were excluded from the statistical analysis, and variance was similar between groups being statistically compared. Significant levels were set at $P \leq 0.05$.

**Reporting summary**. Further information on research design is available in the Nature Research Reporting Summary linked to this article.

## Data availability

Data supporting the findings of this paper are available from the corresponding author upon reasonable request. The source data underlying Figs. 1c, d, 1f, 1h–j, 1l, 1n, 2b–h, 3b-i, 4a, 4c, 4e, 5a–f, 6a–h and Supplementary Figs. 1b–f, 2c, 3b, c, 4b–e, 5b–f, 6c are provided as a Source Data file.

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

## Acknowledgements

We thank Prof. Wenbiao Gan for providing access to a two-photon microscope located in a specific pathogen-free animal facility (New York University Skirball Institute), Drs. Adam Mar and Begona Gamallo (New York University Neuroscience Institute) for the technical assistance with the EthoVision 11.5 software, Prof. Dan Littman (New York University Skirball Institute) for sharing a LSRII equipment, and Prof. Michael Bennett for critical reading of the paper. This work was supported by US National Institutes of Health grants R01GM107469 and R35GM131765 (G.Y.), New York University Dr. Bernard B. Levine postdoctoral fellowship (H.M.S.), and Brazil National Council for Scientific and Technological Development (CNPq) (H.M.S.).

## Author contributions

J.M.G. made the initial behavioral observations in *P2x7r*$^{-/-}$;*Mecp2*[308] mice and initiated the study in collaboration with H.M.S. J.M.G., H.M.S., J.J.L., and G.Y. designed the experiments; J.M.G. performed imaging experiments and animal behavior assays; J.M.G. and H.M.S. performed the flow cytometry experiments; all authors contributed to data analysis; and J.M.G. and G.Y. wrote the paper with input from H.M.S and J.J.L.

## Competing interests

The authors declare no competing interests.
