## [Peer Review File · Nature Communications]

Reviewers' Comments:

Reviewer #1:

Remarks to the Author:

The paper by Garre' and colleagues shows that the microglial receptor P2X7R controls some symptoms in a mouse model of Rett Syndrome (RTT). First, they show an accumulation of the receptor in the Mecp2 mutant mice, then they cross such mice with mice defective of P2X7R. The new generated mice show an improvement of several symptoms and an increased number of spines, together with a reduced accumulation of activated immune-mediating cells at the cortical border zone.

This is an important work which will be of interest of scientists in the field as it suggests a new molecular target for the treatment of Rett Syndrome (RTT). The statistical analysis is adequate. The question this work arises is linked to the progression of the disease and the gender issue. The authors use both genders of mice in this experiment, while it is known that males and females displays symptoms at different ages and have different severity of the disease (linked to several factors among which the X-inactivation in females). How is the activation of P2X7R associated with the onset and progression of the symptoms? It is a necessary factor or the activation or does it correlate with severity of the symptoms?

It would also improve the paper to discuss in what other diseases this receptor has been involved and how these diseases correlate to RTT.

Reviewer #2:

Remarks to the Author:

In this manuscript the authors propose for the first time that myeloid cells expressing cell-surface P2X7R receptors participate mainly to the social deficits of Mecp2 defective mice, modeling Rett syndrome. They also suggest the beneficial effects of targeting these receptors on synaptic plasticity and social behavior, therefore leading the authors to propose P2X7R in peripheral myeloid cells as a novel possible target for the treatment of RTT.

These claims are absolutely novel and relevant for the field and probably even for a broader field linked to autism and/or neurodevelopmental disorders, therefore highlighting the possible relevance of these findings.

Thus, although I find these studies of great importance, I believe that they actually suffer from several technical problems that have to be solved before conveying such an important message. In particular, several are the general major concerns that I will list below.

1) Authors should know that preclinical studies on mice should be performed on littermates (preferentially including pups from at least three different litters), using animals of the same gender and age and randomly assigning the animals to the treatment groups. Further, the investigator should be blind to treatment and group. None of these rules have been respected in the experiments presented along the papers, where both genders have been used in the same group even without specifying how many females or males are present in each group. It is also relevant to notice that the experimental groups are often represented by few animals (4/5) making this concern even bigger. Eventually, it is unacceptable to use a mixed group of animals that are older than 4 months, as well as animals that can be 6 to 12 months old.

2) Most of the results do not specify the number of used animals (I guess we should deduce it from the figures) and/or the animal ages.

3) Most of the statistical analysis is not correct or not well explained. Which Anova are they using in Fig.1? Fig.s 1, 3, 4 and Supp. Fig. 3 use a wrong analysis.

4) The authors claim that BBG does not pass the BBB citing an old paper. However, there are at least three more recent papers claiming the opposite. Further, the experiments proving that it does not pass the BBB are not convincing. This is a relevant issue because they claim that all the

effects are mediated by cells that stay outside of the parenchyma. Therefore, they have to provide convincing data and discuss discrepancy with more recent literature. Eventually, if this would be the case, they should test also a molecule that passes the BB.

5) In the whole manuscript too little experimental information is provided.

In details, the following amelioration are also recommended:

Fig.1 The authors should perform IF studies to really prove the absence of the receptors in the brain; further they should use molecular approaches to demonstrate that pathways downstream of the P2X7R receptors are indeed affected.

Fig. 2 Authors analyze only mutant cells and refer to a publication for the wt distribution. WT controls have to be present in the experiments.

Authors claim that the receptors are increased but they do not observe an effect on their activity. No comment is provided explaining this result.

Fig. 3 There is no text explaining what is M1, M2 and E. I find difficult to understand how such a big effect on synaptic plasticity could lead to such a little and specific behavioral effect. I will suggest including more behavioral analyses and HIC studies to understand the diffusion of the morphological defect in brain and which region is more affected by the treatment and which is unaffected. N=30; what does it mean? 30/group 30 animals for all groups? Why did they choose to analyze frontal cortex instead for example of the prefrontal cortex?

Fig. 4 Which is the age of the donor mice? Authors should perform an IF for the P2X7R receptors on the treated animals.

Supplementary 4, herein authors use A740003 that has never been used before. Why is it? Similarly, now they use motor cortex analysis while they have always analyzed frontal cortex. Eventually, proper controls are missing.

Reviewer #3:

Remarks to the Author:

This MS reports data of a most interesting study, reporting novel results generated by an impressive array of methods. My major criticism relates to the use of BBG as a purely peripheral antagonist of P2X7Rs. The authors cite the paper of Peng et al (PNAS 106:12489, 2009) to substantiate their statement. However, in this paper it was concluded that BBG is a derivative of a commonly used food colour which freely crosses the blood-brain barrier. Admittedly, Peng et al. used a spinal cord lesion model which may lead to the disruption of the blood-spinal cord barrier; however, a high number of other authors (e.g. Parkinson's Disease, Carmo et al., *Neuropharmacology* 81:142, 2014; epilepsy, Rozmer et al., *Cereb. Cort.* 27:3568, 2017) used BBG for ameliorating symptoms of neurodegenerative illnesses. I suggest two sets of experiments to cope with my criticism: (1) to apply one of the JNJ compounds developed by Bhattacharya and his team at the pharmaceutical company Janssen, which is perfectly crossing the blood-brain barrier and (2) to apply BBG intracerebroventricularly.

Minor remarks:

1. P.2, l.4. Please define what you mean by the border of the cerebral cortex (white or grey matter).
2. P.3, 4th para., l.2. Please give an explanation why you used throughout Mecp2-308 mice instead of MeCP2-null mice.
3. P.6, l.1. Instead of "Intermediate-High" the term "intermediate-to-High" might be more correct.
4. P.6, 2nd para., l.2. It is questionable that the permeability increase for fluorescent dyes

develops within seconds to minutes. This is a highly controversial issue discussed repeatedly in the literature and relates to the question whether the electrophysiological findings and the dye uptake experiments signify the same phenomenon or two independent phenomena. The original idea that the reversal potential of the current/voltage curve shows a shift in a medium in which all extracellular Na⁺ was substituted by NMDG⁺ was validly criticized. This appears to be more likely to a redistribution of intra- and extracellular ions. The coupling of P2X7Rs to Panx-1 channels is also questioned.

5. P.7, l.2. Both male and female mice were used (n=30). Does this mean that the distribution among the sexes was 15 males and 15 females?

6. P.7, 2nd para., l.5. I thought that rotarod performance is not affected by deletion of P2X7Rs.

7. P.8, l.1. In which cortical area was abnormal dendritic branching and reduced spine density observed? If this occurs in the prefrontal cortex it is expected to cause depressive-like behaviour.

8. P.11, l.6. The authors state that the low expression of P2X7Rs at microglia is at odds with some data in the literature. This may be due to the low activation grade of microglia in the Mecp2-308 mice (see e.g. Fig. 1n which documents a ramified type of microglia) or the difference between the in vivo and in vitro findings. Under in vitro conditions microglia possess both immunohistochemically and electrophysiologically P2X7Rs. However, many precious publications show massive overlay of P2X7R/Iba1-immunostaining in models of neurodegeneration or CNS damage (see a review cited by the authors themselves; Illes and Verkhratsky, *Neuropharmacology* 104: 62, 2015 or still better in Illes et al., *Neuroscientist* 18:422, 2012).

9. P.12, l.11. As mentioned under item 4, the coupling of P2X7Rs to Panx-1 channels is quite questionable, although ATP may be without doubt released via Panx-1 channels. By the way these are channels, rather than hemichannels, because in contrast to connexins they never link the cytosol of adjacent cells.

Reviewer #4:

Remarks to the Author:

This is a very interesting work addressing the role of peripheral myeloid cells in Rett syndrome. As pointed by the authors, the role of peripheral cells in Rett syndrome is highly controversial, mostly due to lack of immunology knowledge within neuroscience community and due to unwillingness of neuroscientists studying inhibitory neurons in Rett syndrome to accept the fact that additional cells are also involved in this complex disease. Interestingly, Rett girls have numerous pathologies affecting organs beyond the brain, and myeloid cells are the only common denominator to all organs. This work, therefore, is very important and timely. There are a few minor issues that need to be addressed prior to acceptance.

- The authors refer to pial macrophages as cortical and this is incorrect and misleading. Based on the morphology and expressed markers, these are indeed border (pial) macrophages and should not be referred as "cortical". Please revise the text throughout to fix this deficiency.

- It is not clear how the isolation of "cortical macrophages" was performed.

- Have the authors explored other meningeal layers as well as choroid plexus in terms of macrophages?

- There is a typo in the groups listed on top of page 9

- It is surprising that the i.v. injected compound made it to the dura but not to pial macrophages. The authors should perform timeline assessment of this molecule availability to pial macrophages.

- Can the authors protect the Mecp2308 mice by treating them with clodronate liposomes into the CSF? Would simply injection of wild type or P2X7R knockout monocytes improve the outcome?

Finally, would bone marrow transplantation from wild type or P2X7R knockout donors into Mecp2308 mice improve disease outcome? These experiments are not necessary for the current manuscript but would add a great value to the clinical aspect of this paper if performed.

Reviewer #1 (Remarks to the Author):

The paper by Garre' and colleagues shows that the microglial receptor P2X7R controls some symptoms in a mouse model of Rett Syndrome (RTT). First, they show an accumulation of the receptor in the *Mecp2* mutant mice, then they cross such mice with mice defective of P2X7R. The new generated mice show an improvement of several symptoms and an increased number of spines, together with a reduced accumulation of activated immune-mediating cells at the cortical border zone.

This is an important work which will be of interest of scientists in the field as it suggests a new molecular target for the treatment of Rett Syndrome (RTT). The statistical analysis is adequate. The question this work arises is linked to the progression of the disease and the gender issue. The authors use both genders of mice in this experiment, while it is known that males and females displays symptoms at different ages and have different severity of the disease (linked to several factors among which the X-inactivation in females).

We thank the reviewer for the constructive critiques on our study and for pointing out the potential caveats due to the use of mice of both genders. As the reviewer pointed out, male and female *Mecp2* mutant mice display symptoms at different ages and have different severity of the disease, with X-inactivation in *Mecp2*³⁰⁸ females being a major contributing factor. To overcome this issue, we have now revised the manuscript and focused our study on the pathophysiological function of P2X7Rs in *Mecp2*^{308/Y} male mice. The advantages of using *Mecp2*^{308/Y} male mice over other models include that (1) avoiding the variability of behavioral phenotypes associated to X chromosome inactivation; (2) the *Mecp2*^{308/Y} mice allow us studying Rett syndrome phenotypes for longer periods than other mouse models of RTT.

How is the activation of P2X7R associated with the onset and progression of the symptoms? It is a necessary factor or the activation or does it correlate with severity of the symptoms? It would also improve the paper to discuss in what other diseases this receptor has been involved and how these diseases correlate to RTT.

We found that the presence of P2X₇Rs is necessary for *Mecp2*^{308/Y} mice to develop social behavior deficits (*i.e.*, impaired sociability and preference for social novelty). This was demonstrated with a series of genetic and pharmacological experiments. First, we showed that P2X₇R total knockout reduced social defects in *Mecp2*^{308/Y} mice (current **Fig. 4**). Second, using *Mecp2*³⁰⁸ → WT chimeric mice to restrict MECP2 deficiency to bone-marrow (BM) derived leukocytes, we showed that MECP2 deficiency in peripheral leukocytes was sufficient to alter the animals' social behavior, whereas depletion of P2X₇Rs in BM-derived leukocytes prevented these changes (current **Fig. 5**). Third, using pharmacological approaches, we showed that systemic administration of P2X₇R antagonists, such as JNJ-47965567 and BBG, was able to reverse the social behavioral deficits in symptomatic *Mecp2*^{308/Y} mice (current **Fig. 6** and new **Supplementary Fig. 5**). Together, these three lines of evidence indicate that peripheral leukocyte MECP2 deficiency contributes to social behavioral deficits through mechanisms

dependent on P2X₇Rs. We have now revised the manuscript to clarify the above-mentioned data.

It is well established that P2X₇Rs have pro-inflammatory functions. We have now stained the cortex of *Mecp2*^{308/Y} mice with inflammasome activation marker NLRP3 (new **Supplementary Figs. 3 and 5**). We found that *Mecp2*^{308/Y} mice exhibited an increased number of CD206⁺ macrophages in the border of the cerebral cortex (current **Fig. 1a-e**), and a large fraction of these CD206⁺ cells express NLRP3 (new **Supplementary Fig. 3a-c**). Both knocking out (current **Fig. 3**) and inhibiting P2X₇Rs (new **Supplementary Fig. 5a-d**) reduces the number of inflammatory macrophages in the brain of *Mecp2*^{308/Y} mice, suggesting that P2X₇R may contribute to the onset and progression of social behavioral deficits through its impact on brain inflammation. We have included these new data in the Supplementary Information.

In addition to RTT, the protective roles of P2X₇R antagonism have been reported in several other neurological diseases including spinal cord injury (Peng et al, Proc Natl Acad Sci U S A. 2009;106(30):12489-93) and Alzheimer's disease (Martin et al., Mol Psychiatry. 2019, 24(1):108-125). Multiple mechanisms may underlie the protective function of P2X₇R deficiency in these diseases. In the *Mecp2*^{308/Y} mouse model of RTT, we found that P2X₇R activated the NLRP3 inflammasome. NLRP3 activation causes the release of the pro-inflammatory cytokine, IL-1 β , a cytokine that has been associated with neuronal dysfunction. However, in Alzheimer's Disease mouse models, lack of P2X₇Rs did not significantly affect the release of IL-1 β . Instead, P2X₇Rs are shown to play a critical role in A β peptide-mediated chemokine release, which are associated with pathogenic T cell recruitment. Lack of P2X₇Rs reduces amyloid deposition and improves synaptic plasticity and cognitive function. We have now included these in the Discussion.

Reviewer #2 (Remarks to the Author):

In this manuscript the authors propose for the first time that myeloid cells expressing cell-surface P2X₇R receptors participate mainly to the social deficits of *Mecp2* defective mice, modeling Rett syndrome. They also suggest the beneficial effects of targeting these receptors on synaptic plasticity and social behavior, therefore leading the authors to propose P2X₇R in peripheral myeloid cells as a novel possible target for the treatment of RTT.

These claims are absolutely novel and relevant for the field and probably even for a broader field linked to autism and/or neurodevelopmental disorders, therefore highlighting the possible relevance of these findings.

Thus, although I find these studies of great importance, I believe that they actually suffer from several technical problems that have to be solved before conveying such an important message.

In particular, several are the general major concerns that I will list below.

1) Authors should know that preclinical studies on mice should be performed on littermates

(preferentially including pups from at least three different litters), using animals of the same gender and age and randomly assigning the animals to the treatment groups. Further, the investigator should be blind to treatment and group. None of these rules have been respected in the experiments presented along the papers, where both genders have been used in the same group even without specifying how many females or males are present in each group. It is also relevant to notice that the experimental groups are often represented by few animals (4/5) making this concern even bigger. Eventually, it is unacceptable to use a mixed group of animals that are older than 4 months, as well as animals that can be 6 to 12 months old.

We apologize for the confusion due to the inadequate description of experimental details in the previous submission. In the revised manuscript, we have now clarified Methods that all the experiments were performed on littermates, using the animals of similar ages, and all the experimental animals were randomly assigned to the treatment groups. Due to the small litter size (1-2 *Mecp2* mutant males per litter) of *Mecp2* mutant mice, most of our experiments included pups from at least 3 different litters. Throughout the entire study, investigators were blind to the treatments and genotypes during data collection and analysis.

As the reviewer pointed out, animal studies on mice should be performed on animals of the same age and sex. In the original manuscript, most of our experiments (neurological scores, social behavior, flow cytometry analysis, IF, and chimera experiments) were carried out with male mice. We used female mice in a separate behavioral experiment (reciprocal interaction test; previous Fig. 3, now removed) to validate the beneficial effects of P2X₇Rs deficiency observed in *Mecp2*^{308/Y} males. In the revised manuscript, we have decided to focus our study entirely on the male *Mecp2*^{308/Y} mice to avoid the behavioral variability associated to X chromosome inactivation in female *Mecp2*^{308/X} mice, as pointed out by Reviewer 1 (Please also see our response to Reviewer 1). We hope the reviewer is fine with this modification.

In the revised manuscript, we have increased the animal number in behavioral experiments. Because we did not observe significant differences in behavioral phenotypes in young mice between 2 to 3 months of age, we grouped these mice together in social behavioral analysis (current **Fig. 4**). Likewise, we also grouped adult mice that were 4-6 months of ages in social behavioral analysis (current **Fig. 4**).

2) Most of the results do not specify the number of used animals (I guess we should deduce it from the figures) and/or the animal ages.

We have now specified the number and age of animals in the figure legend. In addition, we have now provided a Source Data file that lists all the data in the figures.

3) Most of the statistical analysis is not correct or not well explained. Which Anova are they using in Fig.1? Fig.s 1, 3, 4 and Supp. Fig. 3 use a wrong analysis.

As suggested, we have now specified the details of statistical analysis for each figure in the figure legend. In brief, we used unpaired t test, One-way or Two-way ANOVA followed by Sidak's multiple comparison test in the IF, flow cytometry and social behavioral experiments. Nonparametric tests (Mann-Whitney test, Kruskal-Wallis test) was used in the dendritic spine analysis and neurological score analysis.

4) The authors claim that BBG does not pass the BBB citing an old paper. However, there are at least three more recent papers claiming the opposite. Further, the experiments proving that it does not pass the BBB are not convincing. This is a relevant issue because they claim that all the effects are mediated by cells that stay outside of the parenchyma. Therefore, they have to provide convincing data and discuss discrepancy with more recent literature. Eventually, if this would be the case, they should test also a molecule that passes the BBB.

As the reviewer pointed out, whether BBG passes the BBB or not is a relevant issue. Although BBG's permeability through the BBB is negligible (below the detection limit of 0.01 μM) at the dosage used in our study, we agree with the reviewer that it is not precise to state that BBG does not cross the BBB. In the revised manuscript, we have now removed the statement that BBG does not cross the BBB. Moreover, we have cited recent literature and performed new experiments to measure the concentration of BBG in circulation and brain after systemic administration.

A recent study reported that the brain/plasma ratio of BBG when administered systemically was 0.01 (Fischer et al., 2016, Plos One, DOI:10.1371), which indicates 99% BBG is distributed in plasma, whereas only 1% is detected in brain tissues. After a single dose of 50 mg/kg, BBG in brain reached a concentration of ~ 2 nM, a value that is 200 times below the IC_{50} of mouse $\text{P2X}_7\text{Rs}$ (~ 400 nM, see Fig 2 in Fischer et al., 2016), whereas its concentration in plasma reached micromolar levels ($> \text{IC}_{50}$). In other words, whereas peripheral $\text{P2X}_7\text{Rs}$ will be fully blocked, parenchymal $\text{P2X}_7\text{Rs}$ are not blocked after a single dose of 50 mg/kg BBG. In another study (Diaz-Hernandez et al., 2012, Neurobiology of Aging, Vol 33, 8, 1816-28), Diaz-Hernandez et al., estimated that the brain concentration of BBG after injecting 46 mg/kg BBG every other day for a total of 4 weeks may reach ~ 200 nM, a value still below IC_{50} of mouse $\text{P2X}_7\text{Rs}$.

In the present study, we used a dose of BBG (10 mg/kg) that is 5 times less than the dose (46-50 mg/kg) used in Fischer's and Diaz-Hernandez's studies. Also, the duration of treatment (3-5 days) was substantially shorter than that used in Diaz-Hernandez's study (4 weeks). We have now measured the amount of BBG in the plasma, pial surface, dura and parenchyma, respectively, after i.p. injection of 10 mg/kg/day BBG (new **Supplementary Fig. 6c**). Following 3–5-day treatment, BBG was detected in plasma (~ 30 μM) of both WT and *Mecp2*^{308/Y} mice. BBG was also detected in the duramatter (~ 5 μM) and pial surface of the cortex (~ 12 μM). We did not detect BBG in the cortical parenchyma > 30 μm away from pial surface with our method that has a sensitivity of 0.01 μM , consistent with the previous study (Nedergaard and col., 2009,

PNAS). As a positive control, we created a mouse brain injury using a cortical stab wound, a procedure that has been shown to cause the breakdown of BBB near the injury site. Under the same dosage (10 mg/kg/day, 3-5 days), BBG was detected in the cortical hemisphere ipsilateral, but not contralateral to the injury site. These data indicate very low or none BBG penetrance through the intact BBB at the dosage used in our study. We therefore reasoned that i.p. administration of BBG (10 mg/kg/day, 3-5 days) would primarily inhibit P2X₇Rs in circulating immune cells and macrophages associated with the cortical border (including perivascular spaces, dura and pia mater) (current **Fig. 6**). These results are consistent with our findings of chimeric mice that MECP2 deficiency in peripheral immune cells contributes to social behavioral deficits through bone-marrow derived leukocytes and P2X₇R-dependent mechanisms (current **Fig. 5**). We have now included these new data in the Result of the revised manuscript.

As suggested by the reviewer, we have now added new data using a BBB-permeable P2X₇R antagonist JNJ-47965567 (also suggested by Reviewer 3). After systemic administration, JNJ-47965567 targets P2X₇Rs in circulation and brain equally due to its high brain to plasma ratio (~1). We found that systemic administration of JNJ-47965567 (10 mg/kg, i.p.) for 3-5 days reduced the number of inflammatory myeloid cells in the border of the cerebral cortex (new **Supplementary Fig. 5a-d**) and improved the animals' social behavior (new **Supplementary Fig. 5e,f**). Together, these results indicate the beneficial effects of P2X₇R antagonism in *Mecp2*^{308/Y} mice.

5) In the whole manuscript too little experimental information is provided. In details, the following amelioration are also recommended:

Fig.1 The authors should perform IF studies to really prove the absence of the receptors in the brain; further they should use molecular approaches to demonstrate that pathways downstream of the P2X₇R receptors are indeed affected.

In the present study, we didn't use IF to examine the absence of the P2X₇Rs in the brain. This is because most of commercially available abP2X₇R antibodies not only stain P2X₇Rs, but also unspecifically stain other non-functional P2X₇ splice variants expressed in the brain of *P2x7r*^{-/-} mice (Anderson and Nedergaard, 2006; Niche et al., 2009). At least 4 splice variants have been described for the mouse *P2rx7* gene. P2X₇a is the main variant forming the functional P2X₇Rs in macrophages, whereas P2X₇k splice variant forms the functional channels in naïve and regulatory T cells. Other variants such as P2X₇13b and P2X₇13c are truncated and do not form functional channels. The *P2x7r*^{-/-} mice used in the current study (generated by Pfizer) lack of P2X₇a and P2X₇k but may still express truncated P2X₇ splice variants that can be stained by IF. Indeed, using IF, although we observed the absence of abP2X₇R immuno-reactivity in pial macrophages and microglia, we detected unspecific staining in the cortex (data not shown), as already discussed in the literature (please see the section "Tissue and cell type specific distribution of P2X₇Rs" in Sperlagh and Illes, 2014, Trends in Pharmacological Sciences, and Anderson and Nedergaard, 2006). We hope the reviewer would agree with us that IF is not a

favorable method in detecting P2X₇Rs in the brain with current commercially available antibodies.

Because myeloid cells only express P2X₇a variants which are inserted into the cell surface to form functional channels, we were able to verify the absence of the P2X₇Rs on the surface of (CD45⁺) leukocytes in *P2x7r^{-/-}* mice using flow cytometry (current **Supplemental Fig. 3d**). We have also used PCR to confirm the presence of *P2x7r^{tm1Gab}* mutant allele in all the *P2x7r^{-/-}* mice.

As suggested by the reviewer, we have now provided new data showing that in *Mecp2^{308/Y}* mice, the NLRP3 inflammasome pathway is activated in macrophages in the cortical border via P2X₇R-dependent mechanisms (new **Supplemental Fig. 3a-c**). P2X₇R channel opening by ATP promotes the maturation and release of the pro-inflammatory cytokine IL-1 β through inflammasome-dependent mechanisms. Indeed, we observed a higher number of NLRP3⁺ CD206⁺ macrophages in the cortical border of *Mecp2^{308/Y}* mice as compared to *Mecp2^{308/Y};P2x7r^{-/-}* mice. In the CNS, IL-1 signaling has been associated with neuronal dysfunction (Garber et al., 2018 Nature Immunology 19, 151-161). P2X₇R-induced NLRP3 activation and IL-1 signaling may be coupled to other inflammatory pathways. TNF α has been shown to be involved in cortical synaptic remodeling and is also released upon the activation of P2X₇Rs (Garre et al., 2017, Nature Medicine, 23, 714-722; Barbera-Cremades et al., 2017 Front Immunol 8, 862). Other possible pro-inflammatory pathways downstream of P2X₇R activation include the activation of calcium-dependent PGE₂ release, which has been implicated in behavioral changes associated with sickness (Barbera-Cremades et al., 2012, FASEB J 26, 2951-2962). The inflammatory phenotype driven by P2X₇Rs may also promote IFN γ -mediated immune responses which have been associated with interneuron dysfunction and social deficits (Filiano et al., 2016, Nature 535, 425-429). We have now discussed these pathways in the Discussion.

Fig. 2 Authors analyze only mutant cells and refer to a publication for the wt distribution. WT controls have to be present in the experiments.

We have analyzed the cell surface expression of P2X₇Rs in leukocytes from both WT (current **Fig. 2a-c**) and *Mecp2^{308/Y}* mice (current **Fig. 2d-h**).

Authors claim that the receptors are increased but they do not observe an effect on their activity. No comment is provided explaining this result.

We have now clarified the results on the distribution and activity of P2X₇Rs. P2X₇Rs are abundantly expressed on Ly6C^{high} monocytes and CD206⁺ macrophages in both WT and *Mecp2^{308/Y}* mice (current **Fig. 2a-c**). The surface expression levels of P2X₇Rs on monocytes, macrophages and microglia are not different between WT and *Mecp2^{308/Y}* (current **Fig. 2d-h**). However, comparing *Mecp2^{308/Y}* to WT mice, there is an increased number of CD206⁺ macrophages expressing P2X₇Rs (current **Fig. 1a-k**), suggesting an inflammatory phenotype of

Mecp2^{308/Y} cortex. We have also provided new data showing P2X7Rs are required for the activation of NLRP3 inflammasome pathway in border-associated CD206⁺ macrophages in *Mecp2*^{308/Y} mice (new **Supplementary Fig. 3a-c** and new **Supplementary Fig. 5a-d**). We thank the reviewer for raising this question.

Fig. 3 There is no text explaining what is M1, M2 and E.

We have now explained M1, M2 and E in the figure legend: E, empty cage, M1, Mouse 1 (first stranger); M2, Mouse 2 (second stranger).

I find difficult to understand how such a big effect on synaptic plasticity could lead to such a little and specific behavioral effect. I will suggest including more behavioral analyses and HIC studies to understand the diffusion of the morphological defect in brain and which region is more affected by the treatment and which is unaffected. N=30; what does it mean? 30/group 30 animals for all groups?

In **Fig. 4** of the revised manuscript (previous Fig. 3), we examined the effects of P2X₇R KO on the elimination and formation of dendritic spines over 2 days in the frontal association cortex, one of the cortical regions implicated in the social behavior. We found that rates of spine elimination, but not spine formation, were significantly increased (~5%) in *Mecp2*^{308/Y} mice as compared to WT mice. Knockout of P2X₇Rs in *Mecp2*^{308/Y} mice restored the dynamics of dendritic spines in the frontal association cortex (current **Fig. 3**) and attenuated the social behavioral deficits in *Mecp2*^{308/Y} mice (current **Fig. 4**). Together with the reduction of CD206⁺ macrophages in the cortical border of *Mecp2*^{308/Y}; *P2x7*^{-/-} mice, these results indicate the pathophysiological role of P2X₇Rs in RTT mice.

We agree with the reviewer that it is important to determine the morphological defects in other brain regions involved in the social behavior (e.g., hippocampus, prefrontal cortex). Unfortunately, because hippocampal and prefrontal synapses are not accessible using a non-invasive in vivo imaging approach, they could only be examined in vivo with the implantation of a GRIN lens or prism. Moreover, HIC studies cannot be used for the analysis of dendritic spine dynamics (i.e., spine elimination and spine formation). For the technical reasons, we hope the reviewer would agree with us that it would be more appropriate to develop better tools to investigate the effects of *Mecp2* mutation and P2X₇R KO in other brain regions in a future study.

We have now increased the animal number in behavioral experiments and provided the n number for each group in the figure legend.

Why did they choose to analyze frontal cortex instead for example of the prefrontal cortex?

Social behavior involves multiple brain regions including, but not limited to, ventral hippocampus, the medial prefrontal cortex and the frontal association cortex (Phillips et al., *elife* 2019, 8, 44182; Shu et al., *Neural Plast* 2017, 6207873). We chose the frontal association cortex for *in vivo* dendritic spine imaging because this cortical region is right under the skull surface and is optically accessible using a transcranial two-photon imaging approach. It is technically challenging to image mouse hippocampus and prefrontal cortex while keeping the meningeal compartment intact.

Fig. 4 Which is the age of the donor mice? Authors should perform an IF for the P2X7R receptors on the treated animals.

The donor mice are of 4 months of age. We have used PCR to verify the presence or absence of P2X7R mutant alleles in bone marrow-derived leukocytes in all chimeric mice used.

Supplementary 4, herein authors use A740003 that has never been used before. Why is it? Similarly, now they use motor cortex analysis while they have always analyzed frontal cortex. Eventually, proper controls are missing.

We have now removed A740003 data from the manuscript. The revised manuscript focused on the effects of the Brilliant Blue G (BBG) and JNJ-47965567 treatment on the social behavior and cortical inflammation in *Mecp2*^{308/Y} mice (current **Fig. 6** and new **Supplemental Fig. 5**). We hope the reviewer is fine with this modification.

Reviewer #3 (Remarks to the Author):

This MS reports data of a most interesting study, reporting novel results generated by an impressive array of methods. My major criticism relates to the use of BBG as a purely peripheral antagonist of P2X7Rs. The authors cite the paper of Peng et al (PNAS 106:12489, 2009) to substantiate their statement. However, in this paper it was concluded that BBG is a derivative of a commonly used food colour which freely crosses the blood-brain barrier. Admittedly, Peng et al. used a spinal cord lesion model which may lead to the disruption of the blood-spinal cord barrier; however, a high number of other authors (e.g. Parkinson's Disease, Carmo et al., *Neuropharmacology* 81:142, 2014; epilepsy, Rozmer et al., *Cereb. Cort.* 27:3568, 2017) used BBG for ameliorating symptoms of neurodegenerative illnesses.

Thanks for the positive view about our MS and for bringing up this question. We have now performed new experiments to measure the amount of BBG in circulation and brain after systemic administration. We have now clarified in the manuscript that although BBG largely stays in the periphery after systemic administration, it can be detected in the border region of the cortex at sufficient concentrations (~2 times higher than the IC₅₀ of mouse P2X₇Rs) to block P2X7Rs. We did not detect BBG in the cortical parenchyma >30 μm away from the pial surface (Please also see our response to Reviewer 2).

In the revised manuscript, we described the BBG experiments as following:

“Next, we tested whether the beneficial effects of blocking P2X₇Rs in *Mecp2*^{308/Y} mice can be achieved by primarily inhibiting P2X₇Rs in the periphery. Brilliant Blue G is a U.S. Food and Drug Administration-approved drug and has been shown to block P2X₇R-mediated increase in membrane permeability and macroscopic currents in cell cultures and spinal cord slices. Previous studies have shown that BBG has low penetrance to the CNS parenchyma (brain to plasma ratio ~0.01). After i.p. injection at 50 mg/kg, the estimated concentration of BBG in the brain is about 0.002 μM, which is 200 times lower than the IC₅₀ of mouse P2X₇Rs. Indeed, we found that i.p. injection of 10 mg/kg BBG colored plasma samples, abdominal tissues and highly vascularized duramater, but did not stain the cerebral cortex (excluding meninges) (current **Supplementary Fig. 6a, b**). Following 3–5-day treatment, BBG was detected in plasma (~30 μM) of both WT and *Mecp2*^{308/Y} mice (new **Supplementary Fig. 6c**), using a method with a sensitivity of 0.01 μM. Similarly, BBG was also detected in the superficial layer of the cortex up to 30 μm below pial surface (~12 μM) and duramater (~5 μM). Brilliant Blue G was not detected in the cortical parenchyma beyond 30 μm from the pial surface (new **Supplementary Fig. 6c**). We therefore reasoned that i.p. administration of BBG would primarily inhibit P2X₇Rs in circulating immune cells and macrophages associated with the cortical border (including perivascular spaces, dura and piamater). Notably, we found that BBG treatment for 3-5 days significantly reduced the relative numbers of LY6C^{High} monocytes and CD206^{High}MHCII^{Low} and CD206^{Low}MHCII^{High} macrophages in the cortex of *Mecp2*^{308/Y} mice, with no significant effects on the microglial fraction (**Fig. 6a-d**).”

I suggest two sets of experiments to cope with my criticism: (1) to apply one of the JNJ compounds developed by Bhattacharya and his team at the pharmaceutical company Janssen, which is perfectly crossing the blood-brain barrier and (2) to apply BBG intracerebroventricularly.

As suggested by the reviewer, we have now performed new behavioral experiments using JNJ-47965567 compounds.

“JNJ-47965567 is a BBB-permeable P2X₇R antagonist. After systemic administration, JNJ-47965567 targets P2X₇Rs in circulation and brain equally due to its high brain to plasma ratio (~1). Consistent with the findings of P2X₇R knockout mice, we found that systemic administration of JNJ-47965567 (10 mg/kg/day, i.p.) for 3 days reduced the number of inflammatory myeloid cells in the border of cerebral cortex (new **Supplementary Fig. 5a-d**) and improved the animals' social behavior (new **Supplementary Fig. 5e, f**), indicating the beneficial effects of P2X₇R antagonism in *Mecp2*^{308/Y} mice”

We thank the reviewer for suggesting the experiments of applying BBG intracerebroventricularly. For the following reasons, we did not perform these experiments.

First, in the new **Supplementary Fig. 5**, we have now shown that systemic administration of JNJ-47965567 reduced social deficits in *Mecp2*^{308/Y} mice. In the current **Fig. 6**, we showed that systemic administration of BBG reverses social behavioral deficits in *Mecp2*^{308/Y} mice. In the new **Supplementary Fig. 5**, we have now provided new data that systemically administered BBG can be detected in the border, but not deep parenchymal region of the cortex. Intraperitoneally injected BBG would, therefore, primarily inhibit P2X₇Rs in circulating immune cells, border associated macrophages, but not microglia. Together, these results indicate that the beneficial effects of P2X₇R antagonism in *Mecp2*^{308/Y} mice are largely mediated by peripheral immune cells and border associated macrophages. Indeed, our data show that both JNJ-47965567 and BBG treatments reduced the number of inflammatory macrophages in the cortical border (new **Supplementary Fig. 5** and **Fig 6**). These pharmacology experiments are also consistent with our genetic findings that lack of P2X₇Rs in leukocytes prevented *Mecp2*^{308/Y} leukocytes-induced social behavioral deficits (current **Fig. 5**).

Second, given our new data on the distribution of BBG after ip injection (new **Supplementary Fig. 6c**), we think that experiments with intracerebroventricular injection of BBG may not provide additional information because intracerebroventricularly injected BBG will target not only parenchyma microglia, but also monocytes/macrophages located in the meninges and perivascular spaces which are connected through the cerebral spinal fluid circulation.

Third, intracerebroventricular injection is an invasive procedure which may activate meningeal macrophages and microglia and confound the behavioral analysis.

Minor remarks:

1. P.2, l.4. Please define what you mean by the border of the cerebral cortex (white or grey matter).

We have now clarified the definition of cortical border. In the current study, we quantified the density of macrophages in the superficial layer of the cortex within 30 μm from pial surface, including pial mater. The analysis of cortical parenchyma was performed from brain tissues >30 μm away from pial surface.

2. P.3, 4th para., l.2. Please give an explanation why you used throughout *Mecp2*-308 mice instead of *MeCP2*-null mice.

We have now explained in the Introduction that “In this study, we investigated the pathophysiological role of P2X₇Rs in mice expressing a truncated form of MeCP2 (*Mecp2*^{308/Y}, also known as *Mecp2*^{tm1Hzo}), which is a well-established model to study RTT phenotypes in male mice, avoiding the variability of behavioral outcomes linked to X-chromosome inactivation”.

3. P.6, l.1. Instead of “Intermediate-High” the term “intermediate-to-High” might be more correct.

We have now corrected it.

4. P.6, 2nd para., l.2. It is questionable that the permeability increase for fluorescent dyes develops within seconds to minutes. This is a highly controversial issue discussed repeatedly in the literature and relates to the question whether the electrophysiological findings and the dye uptake experiments signify the same phenomenon or two independent phenomena. The original idea that the reversal potential of the current/voltage curve shows a shift in a medium in which all extracellular Na⁺ was substituted by NMDG⁺ was validly criticized. This appears to be more likely to a redistribution of intra- and extracellular ions. The coupling of P2X7Rs to Panx-1 channels is also questioned.

In the revised manuscript, we have clarified that P2X₇Rs mediate the increase in membrane permeability to large molecules (≤900 DA) and fluorescent tracers after ATP treatment.

5. P.7, l.2. Both male and female mice were used (n=30). Does this mean that the distribution among the sexes was 15 males and 15 females?

The revised manuscript focused on the male mice only. We have now specified the animal number in each group in the figure legend.

6. P.7, 2nd para., l.5. I thought that rotarod performance is not affected by deletion of P2X7Rs.

As the reviewer pointed out correctly, rotarod performance is not affected by P2X₇R deletion. We have now clarified the manuscript: "When assessed with intensive rotarod training (40 trials), *Mecp2*^{308/Y} mice showed lower performance relative to WT mice. However, there were no differences in rotarod performances between *P2x7r*^{-/-}, *Mecp2*³⁰⁸ and *Mecp2*^{308/Y}; *P2x7r*^{-/-} mice (**Supplementary Fig. 4e**)."

7. P.8, l.1. In which cortical area was abnormal dendritic branching and reduced spine density observed? If this occurs in the prefrontal cortex it is expected to cause depressive-like behaviour.

In the present study, we observed increased dendritic spine elimination in the frontal association cortex of *Mecp2*^{308/Y} mice. This is consistent with previous findings of abnormal dendritic branching and reduced spine density in multiple brain regions of RTT mouse models and patients (Xu et al., *Front Neuroanat.* 2014; 8:97). As the reviewer pointed out correctly, the prefrontal cortex is also involved in the depressive-like behavior. Indeed, loss of MeCP2 has been shown to cause depression phenotypes in mice (Philippe et al, 2018, *Scientific Reports* 8: 5788). Since the present study focused on the impact of *Mecp2* deficiency on the social behavior, we did not assess depression-like behavior in these mice.

8. P.11, I.6. The authors state that the low expression of P2X₇Rs at microglia is at odds with some data in the literature. This may be due to the low activation grade of microglia in the *Mecp2*-308 mice (see e.g. Fig. 1n which documents a ramified type of microglia) or the difference between the in vivo and in vitro findings. Under in vitro conditions microglia possess both immunohistochemically and electrophysiologically P2X₇Rs. However, many precious publications show massive overlay of P2X₇R/Iba1-immunostaining in models of neurodegeneration or CNS damage (see a review cited by the authors themselves; Illes and Verkhratsky, *Neuropharmacology* 104: 62, 2015 or still better in Illes et al., *Neuroscientist* 18:422, 2012).

We agree with the reviewer that the low expression of P2X₇Rs on microglia could be due to the low activation grade of microglia in *Mecp2*^{308/Y} mice. Indeed, in the cortex of *Mecp2*^{308/Y} mice, we did not observe any morphological signs of microglial activation (current **Fig. 1l, m**), although the density of microglia was reduced by 10% as compared to that in WT mice (current **Fig. 1c**). There were no differences in the levels of myeloid activation markers MHCII and CD11c in microglia between WT and *Mecp2*^{308/Y} mice (current **Fig. 1n, o**). We have now revised the discussion: “Such discrepancy can be attributed to less microglial activation in the cortex of *Mecp2*^{308/Y} mice, as well as our excluding macrophages in the flow cytometry analysis”. We thank the reviewer for the suggestion.

9. P.12, I.11. As mentioned under item 4, the coupling of P2X₇Rs to Panx-1 channels is quite questionable, although ATP may be without doubt released via Panx-1 channels. By the way these are channels, rather than hemichannels, because in contrast to connexins they never link the cytosol of adjacent cells.

As the reviewer point out correctly, pannexin channels comprise one pathway for ATP release. Secreted ATP may activate P2X₇Rs and promote further ATP release. Pannexin-1 channels are also referred to as Pannexin-1 hemichannels. In fact, mouse and human pannexins are orthologs of innexins, the proteins forming gap junctions in insects. The name pannexin was coined by Panchin and denotes that these innexin-like proteins are broadly found in both invertebrates and vertebrates (pan=everywhere). Although gap junctions formed of pannexins have not been described in vertebrates in vivo yet, dye and electrical coupling have been reported in cell lines transfected with mammalian pannexins. We used (hemi)channel in the discussion to support two common terminologies found in the literature: “pannexin channel” and “pannexin hemichannel”. Although pannexon is the “unquestionable” terminology, it is used less frequently in the field of gap junctions, likely to avoid outsider confusion with the term pannexin. We have now revised the discussion on P2X₇Rs and Panx-1 channels. “Furthermore, activation of P2X₇Rs leads to the opening of accessory channels such as Pannexin1 (hemi)channels (Px1 HCs), which mediates ATP release, supporting P2X₇R activation within the inflammatory microenvironment. The cooperation between P2X₇Rs and Px1 HCs and its implications in physiology and pathology have been extensively studied in astrocytes, but it is still unclear

whether, and if so how, this cooperation exists in monocytes and macrophages in the CNS.” We hope reviewer is fine with these modifications.

Reviewer #4 (Remarks to the Author):

This is a very interesting work addressing the role of peripheral myeloid cells in Rett syndrome. As pointed by the authors, the role of peripheral cells in Rett syndrome is highly controversial, mostly due to lack of immunology knowledge within neuroscience community and due to unwillingness of neuroscientists studying inhibitory neurons in Rett syndrome to accept the fact that additional cells are also involved in this complex disease. Interestingly, Rett girls have numerous pathologies affecting organs beyond the brain, and myeloid cells are the only common denominator to all organs. This work, therefore, is very important and timely. There are a few minor issues that need to be addressed prior to acceptance.

- The authors refer to pial macrophages as cortical and this is incorrect and misleading. Based on the morphology and expressed markers, these are indeed border (pial) macrophages and should not be referred as “cortical”. Please revise the text throughout to fix this deficiency.

In the revised manuscript, we have now removed the term “cortical macrophages”. We also clarified that these CD206⁺ macrophages were observed in the border of the cerebral cortex and perivascular spaces, but not in the deep cortical parenchyma.

- It is not clear how the isolation of “cortical macrophages” was performed.

We characterized macrophages in the brain by performing immunohistochemistry and flow cytometry. In the revised Method, we have clarified that IHC and flow were performed from cortical tissues with leptomeninges intact but duramater removed.

- Have the authors explored other meningeal layers as well as choroid plexus in terms of macrophages?

In the present study, we did not examine immune cells in choroid plexus, as well as in other meningeal layers. We chose to focus on pial macrophages (*i.e.*, leptomeningeal macrophages) because (1) these macrophages are located close to the superficial layer of cortex in which we have observed abnormal dendritic spine plasticity in the apical dendrites of layer 5 pyramidal neurons; (2) leptomeningeal macrophages express high levels of P2X₇Rs, in contrast to dura macrophages which seem to express lower levels (Van Hove et al., 2019). We have cited several recent studies characterizing meningeal and perivascular macrophages in CNS tissues (Goldman et al., Nature Immunology 2016; Mrdjen et al., Immunity 2018; Van Hove et al., Nature Neuroscience 2019).

- There is a typo in the groups listed on top of page 9

Corrected.

- It is surprising that the i.v. injected compound made it to the dura but not to pial macrophages. The authors should perform timeline assessment of this molecule availability to pial macrophages.

In the revised manuscript, we have now provided direct evidence that systemically administered BBG made it to both dura and pial mater, but at much lower concentration than that in plasma (new **Supplementary Fig. 6c**). Specifically, we have measured the concentration of BBG in plasma, dura, pial border, and cortical parenchyma, respectively, after i.p. injection of 10 mg/kg/day BBG. Following 3–5-day treatment, BBG was detected in plasma (~30 μM) of both WT and *Mecp2*^{308/Y} mice. BBG was also detected in the superficial layer of the cortex within 30 μm below pial surface (~12 μM) and duramatter (~5 μM). BBG concentrations ranging 5-15 μM are sufficient to fully block mouse P2X₇Rs (IC₅₀ of P2X₇R is 0.4 μM , please also see our response to reviewer 2). BBG was not detected in the cortical parenchyma 30 μm beyond the pial surface, using our method with a sensitivity of 0.01 μM . Together, these new experiments indicated that BBG can reach pial macrophages following systemic administration.

- Can the authors protect the *Mecp2*³⁰⁸ mice by treating them with clodronate liposomes into the CSF? Would simply injection of wild type or P2X₇R knockout monocytes improve the outcome? Finally, would bone marrow transplantation from wild type or P2X₇R knockout donors into *Mecp2*³⁰⁸ mice improve disease outcome? These experiments are not necessary for the current manuscript but would add a great value to the clinical aspect of this paper if perfomed.

We thank the reviewer for proposing the experiments with in vivo macrophage depletion, adoptive cell transfer and *Mecp2*^{308/Y};*P2x7r*^{-/-} → *Mecp2*^{308/Y} chimeras, which would likely add additional value to the current study. However, as the reviewer may understand, these new experiments will require a large number of *Mecp2*^{308/Y} and *Mecp2*^{308/Y};*P2x7r*^{-/-} mice and the generation of chimeric mice will take months. At the moment, we don't have enough number of animals to perform these experiments. Further studies would be needed to investigate the roles of peripheral monocytes/macrophages in reversing the neurological dysfunction in RTT. We have added this point in the Discussion section.

Reviewers' Comments:

Reviewer #1:

Remarks to the Author:

The revisions improved the manuscript that is now suitable for publication

Reviewer #2:

None

Reviewer #3:

Remarks to the Author:

The authors perfectly answered the questions put up by me.

Reviewer #4:

Remarks to the Author:

All of my concerns have been addressed

Response to reviewers' comments

Reviewer #1 (Remarks to the Author):

The revisions improved the manuscript that is now suitable for publication.

Thank you.

Reviewer #3 (Remarks to the Author):

The authors perfectly answered the questions put up by me.

Thank you.

Reviewer #4 (Remarks to the Author):

All of my concerns have been addressed.

Thank you.